# DYNAMIC NEGATIVE GUIDANCE OF DIFFUSION MODELS

**Felix Koulischer**[1,*]        **Johannes Deleu**[1]        **Gabriel Raya**[2]
**Thomas Demeester**[1,†]        **Luca Ambrogioni**[3,†]

[1] IDLab, Ghent University        [2] JADS, Tilburg University
[3] Donders Institute for Brain Cognition and Behaviour, Radboud University

## ABSTRACT

Negative Prompting (NP) is widely utilized in diffusion models, particularly in text-to-image applications, to prevent the generation of undesired features. In this paper, we show that conventional NP is limited by the assumption of a constant guidance scale, which may lead to highly suboptimal results, or even complete failure, due to the non-stationarity and state-dependence of the reverse process. Based on this analysis, we derive a principled technique called *Dynamic Negative Guidance*, which relies on a near-optimal time and state dependent modulation of the guidance without requiring additional training. Unlike NP, negative guidance requires estimating the posterior class probability during the denoising process, which is achieved with limited additional computational overhead by tracking the discrete Markov Chain during the generative process. We evaluate the performance of DNG class-removal on MNIST and CIFAR10, where we show that DNG leads to higher safety, preservation of class balance and image quality when compared with baseline methods. Furthermore, we show that it is possible to use DNG with Stable Diffusion to obtain more accurate and less invasive guidance than NP. Our implementation is available at `https://github.com/FelixKoulischer/Dynamic-Negative-Guidance.git`

## 1 INTRODUCTION

Since first proposed as generative models (Sohl-Dickstein et al., 2015; Ho et al., 2020; Karras et al., 2022; Song et al., 2021), Diffusion models (DMs) have surpassed previous generative models by large margins on a wide range of tasks, including Text-to-Image (T2I) generation (Rombach et al., 2022), audio synthesis (Kong et al., 2021), video synthesis (Bar-Tal et al., 2024), and protein design (Watson et al., 2023). One of the key features behind their success is the simplicity with which these models can be controlled during generation. This process, referred to as guidance (Dhariwal & Nichol, 2021; Ho & Salimans, 2021), allows users to sample from a conditional distribution. In modern T2I DMs the guidance is performed through joint text-image latent representations such as CLIP for Stable Diffusion (Radford et al., 2021; Rombach et al., 2022; Podell et al., 2024). The language model behind such a joint embedding is however quite poor and struggles to understand basic textual operations such as negations (Parcalabescu et al., 2022; Singh et al., 2024; Quantmeyer et al., 2024). As an example, when prompting Stable Diffusion with the prompt: "A gentleman with a cane from the 1800s without a mustache", it is almost guaranteed that the model will produce an image of a man with a mustache. To resolve this, widespread diffusion models such as Midjourney or Stable Diffusion rely on a second prompt line that contains a so-called *negative prompt* (Gandikota et al., 2023a; Ban et al., 2024; Armandpour et al., 2023; Schramowski et al., 2023; Du et al., 2020). In the example mentioned above, this line would contain the prompt: "A mustache". This prompt is then used similarly to the positive prompt, the only difference being that instead of pushing the model *towards* the condition, the model pushes *away* from it. This is called Negative Prompting (NP). Negative prompting is commonly accepted in the community, and all publicly available models

---

[*]Corresponding author: *felix.koulischer@ugent.be*
[†] Joint Senior Authors

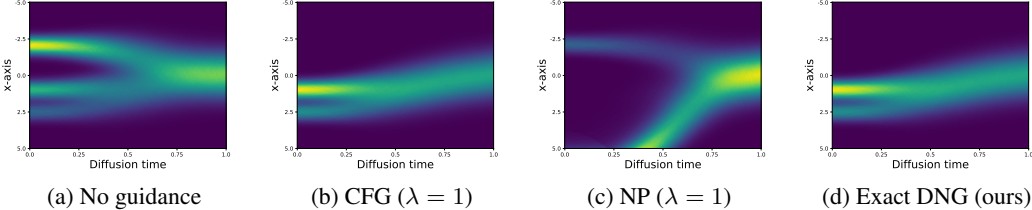

| (a) No guidance | (b) CFG ($\lambda = 1$) | (c) NP ($\lambda = 1$) | (d) Exact DNG (ours) |

Figure 1: Comparison between CFG, NP and DNG using the exact posterior (all used with $\lambda = 1$). DNG with the exact posterior is equivalent to CFG, while NP fails to sample the target distribution.

rely on some form of NP. Its theoretical aspects remain however understudied (Armandpour et al., 2023; Ban et al., 2024). Mathematically, NP is straightforward, as it corresponds to the well-studied Classifier-Free Guidance (CFG) scheme (Ho & Salimans, 2021) with a negative guidance scale. However, while effective in practice, such a simple change of sign is fundamentally flawed. CFG can be seen as an attractive vector field that aims to guide the model towards the correctly conditioned outputs. In particular, the further away the noised image is from the desired condition, the stronger the guidance field is. When designing a guidance scheme that *averts* the denoising process towards an output from an undesired region, two properties must be met. First, the corresponding repulsive field needs to be directed away from the undesired conditional, i.e. a change of sign is required, which NP satisfies by construction. In addition, the repulsive vector field should be the strongest close to the undesired regions, such that the negative prompt has no impact at all if the progressive denoising process remains far from any of the undesired regions. This property is missing from the traditional NP scheme. The flaws of NP are particularly apparent in one dimension, in which NP completely fails to sample the desired conditional distribution. This is shown in Fig. 1, in which the different guidance schemes are compared to each other. Another key flaw of NP is the static nature of the guidance scheme. It was recently shown that the generative denoising process is far from being uniform in time, as it is instead demarcated by brief symmetry breaking events corresponding to 'generative decisions' (Raya & Ambrogioni, 2024; Biroli et al., 2024; Ambrogioni, 2024a; Sclocchi et al., 2024; Li & Chen, 2024). Ideally, it is during those key bifurcation moments that guidance should be active, since outside those semantic decision regions guidance only harms the diversity of the model (Kynkäänniemi et al., 2024; Sclocchi et al., 2024).

To resolve these issues, we propose a novel theoretically grounded dynamic negative guidance scheme, coined **D**ynamic **N**egative **G**uidance (DNG), which in the static limit reduces to the widespread NP scheme. Contrary to NP, our approach requires estimating the posterior, which we achieve by tracking the discrete Markov chain during the denoising process. The dynamic guidance scale is proportional to the posterior, which causes the guidance to vanish whenever the posterior probability of the "forbidden class" is low. This is illustrated in Fig. 5, in which the guidance scale of images generated using Stable Diffusion (Rombach et al., 2022) is plotted as a function of the denoising time next to illustrative diffusion trajectories. When using a negative prompt related to the positive prompt, the guidance is active, while when using a semantically unrelated negative prompt, the negative guidance scale drops to zero. Our dynamic approach can self-regulate and even deactivate itself in the case that the negative prompt is completely unrelated to the positive prompt, which is something which NP is incapable of doing. The code associated with this paper will be made publicly available upon acceptance.

## 2 RELATED WORK

To introduce our dynamic negative guidance scheme, the working principles behind Denoising Diffusion Probabilistic Models (DDPMs), Classifier-Free-Guidance (CFG) and Negative Prompting (NP) are required. These are summarized in sections 2.1 and 2.2. In section 2.3, the related topic of concept erasure in diffusion models is briefly discussed.

## 2.1 Denoising Diffusion Probabilistic Models (DDPM)

Score based diffusion models (Sohl-Dickstein et al., 2015; Ho et al., 2020; Song et al., 2021) are progressive denoisers, that learn to invert a forward noising schedule. In the case of variance preserving DDPM (Ho et al., 2020), this forward noising process, iteratively adds Gaussian noise to a clean data distribution. The underlying data distribution $q(\boldsymbol{x}, 0)$ is progressively noised until a standard normal Gaussian is obtained, i.e. until after $T$ steps $q(\boldsymbol{x}, T) \sim \mathcal{N}(\boldsymbol{x}; \boldsymbol{0}, \boldsymbol{I})$. For any forward noising schedule $\{\beta_t\}_{t=1}^T$, whereby $\boldsymbol{x}_{t+1} = \sqrt{1 - \beta_t}\boldsymbol{x}_t + \sqrt{\beta_t}\boldsymbol{\epsilon}_t$ (with $\boldsymbol{\epsilon}_t \sim \mathcal{N}(\boldsymbol{\epsilon}_t; \boldsymbol{0}, \boldsymbol{I})$)), the forward process can be directly sampled at any step $t$, according to $q(\boldsymbol{x}_t|\boldsymbol{x}_0) \sim \mathcal{N}(\boldsymbol{x}_t; \sqrt{\bar{\alpha}_t}\boldsymbol{x}_0, (1 - \bar{\alpha}_t)\boldsymbol{I})$, with $\bar{\alpha}_t = \prod_{i=0}^t (1 - \beta_i)$. The goal is to learn a score based model, defined by $\boldsymbol{s}_\theta(\boldsymbol{x}, t) = \nabla_{\boldsymbol{x}} \log p_t(\boldsymbol{x}; \theta)$, that is able to iteratively reverse the forward process. The reverse process can be decomposed into a Markov chain

$$p(\boldsymbol{x}_{t:T}; \theta) = p(\boldsymbol{x}_T) \prod_{i=t+1}^T p_i(\boldsymbol{x}_{i-1}|\boldsymbol{x}_i; \theta) \tag{1}$$

Importantly, each step of the Markov chain is by construction approximately Gaussian $p_t(\boldsymbol{x}_{t-1}|\boldsymbol{x}_t; \theta) \sim \mathcal{N}(\boldsymbol{x}_{t-1}; \boldsymbol{\mu}_\theta(\boldsymbol{x}_t), \sigma_t^2 \boldsymbol{I})$. Instead of modeling the mean, $\boldsymbol{\mu}_{\boldsymbol{\theta},\boldsymbol{t}}$, it is common to instead predict the noise present at any diffusion stage $\boldsymbol{\epsilon}_{\boldsymbol{\theta},\boldsymbol{t}}$. The two are linked through a linear combination

$$\boldsymbol{\mu}_{\boldsymbol{\theta},t-1} = \frac{1}{\sqrt{1 - \beta_t}}\Big(\boldsymbol{x}_t - \frac{\beta_t}{\sqrt{1 - \bar{\alpha}_t}}\boldsymbol{\epsilon}_{\boldsymbol{\theta},t-1}\Big) \tag{2}$$

The model can then be trained using as optimization objective

$$\boldsymbol{\theta}^\star = \arg\min_{\boldsymbol{\theta}} \mathbb{E}_{t \sim \mathcal{U}(0,T), \boldsymbol{x}_0, \boldsymbol{\epsilon}} \Big[\|\boldsymbol{\epsilon}_{\boldsymbol{\theta},t}(\sqrt{\bar{\alpha}_t}\boldsymbol{x}_0 + \sqrt{1 - \bar{\alpha}_t}\boldsymbol{\epsilon}) - \boldsymbol{\epsilon}\|^2\Big].$$

It should also be noted that the error prediction $\boldsymbol{\epsilon}_\theta$ is proportional to the score function, more specifically $\boldsymbol{s}_{\boldsymbol{\theta},t} = \nabla_{\boldsymbol{x}} \log p_t(\boldsymbol{x}_t; \theta) = -\boldsymbol{\epsilon}_{\boldsymbol{\theta},t}(\boldsymbol{x})/\sqrt{1 - \bar{\alpha}_t}$.

## 2.2 Classifier-Free Guidance and Negative Prompting

Most practical applications require *conditional generation* (Dhariwal & Nichol, 2021; Ho & Salimans, 2021; Schramowski et al., 2023). The first approach to achieve this, introduced by Dhariwal & Nichol (2021), is called *classifier guidance* (CG). They consider a classifier $p(\boldsymbol{c}|\boldsymbol{x})$ that quantifies to what extent any condition $\boldsymbol{c}$ is met by the object $\boldsymbol{x}$ under generation, such as a desired class label or adherence to a textual description. The joint probability can be constructed from an unconditional model $p(\boldsymbol{x})$ combined with an external classifier $p(\boldsymbol{c}|\boldsymbol{x})$, i.e., $p(\boldsymbol{x}, \boldsymbol{c}) = p(\boldsymbol{x})p(\boldsymbol{c}|\boldsymbol{x})$. By further sharpening the posterior $p(\boldsymbol{c}|\boldsymbol{x})$ using a positive exponent $\lambda > 1$, i.e. sampling from $p(\boldsymbol{x}, \boldsymbol{c}) \propto p(\boldsymbol{x})p(\boldsymbol{c}|\boldsymbol{x})^\lambda$, they were able to obtain superior results[1]. The exponent $\lambda$ is called the guidance scale. The main impact of this exponent is to make sure that the condition $\boldsymbol{c}$ is more closely respected, which comes at the cost of diversity (Ho & Salimans, 2021; Kynkäänniemi et al., 2024). Like all score-based models, DMs do not directly model the data distribution. Instead, they restrict themselves to modeling the so-called score-function, the gradient of the log-likelihood. For CG this results in:

$$\nabla_{\boldsymbol{x}} \log p_t(\boldsymbol{x}|\boldsymbol{c}) = \nabla_{\boldsymbol{x}} \log p_t(\boldsymbol{x}) + \lambda \nabla_{\boldsymbol{x}} \log p_t(\boldsymbol{c}|\boldsymbol{x}) \tag{3}$$

The problem behind such an approach is that for most of the diffusion process, the images are strongly noised, rendering pretrained classifiers ineffective. Instead, a new time-dependent classifier would have to be trained to recognize images during the denoising process, which is not only costly but also suboptimal. This is why Ho & Salimans (2021) proposed to directly train the joint distribution $p_t(\boldsymbol{x}, \boldsymbol{c})$ using pairs of textual and visual representations, such as LAION (Schuhmann et al., 2022). Their key insight was to keep the guidance scale $\lambda$ and reuse Bayes rule again to rewrite the posterior $p_t(\boldsymbol{c}|\boldsymbol{x}) = p_t(\boldsymbol{x}, \boldsymbol{c})/p_t(\boldsymbol{x})$. In score notation, this results in:

$$\nabla_{\boldsymbol{x}} \log p_t(\boldsymbol{x}|\boldsymbol{c}) = \nabla_{\boldsymbol{x}} \log p_t(\boldsymbol{x}) + \lambda\big(\nabla_{\boldsymbol{x}} \log p_t(\boldsymbol{x}, \boldsymbol{c}) - \nabla_{\boldsymbol{x}} \log p_t(\boldsymbol{x})\big) \tag{4}$$

Writing the unconditional score as $\nabla_{\boldsymbol{x}} \log p_t(\boldsymbol{x}) = \boldsymbol{s}_\theta$, and the positively conditioned score as $\nabla_{\boldsymbol{x}} \log p_t(\boldsymbol{x}, \boldsymbol{c}) = \boldsymbol{s}_{\theta,c_+}$ yields the well-known CFG equation:

$$\boldsymbol{s}_{CFG} = \boldsymbol{s}_\theta + \lambda\big(\boldsymbol{s}_{\theta,c_+} - \boldsymbol{s}_\theta\big) \tag{5}$$

---

[1]Strictly speaking, expressions such as $p(\boldsymbol{x})p(\boldsymbol{c}|\boldsymbol{x})^\lambda = p(\boldsymbol{x}, \boldsymbol{c})$ are notationally inconsistent for $\lambda$ different than one, but to remain consistent with literature, the loose notation is kept.

For image generation conditioned on textual prompts, $s_\theta$ is obtained by providing the trained joint model with an empty prompt. It soon became clear that a similar approach allows guiding the model away from undesired prompts, by reversing the sign of the guidance scale in the CFG equation (5), leading to the Negative Prompting (NP) process:

$$s_{NP} = s_\theta - \lambda\big(s_{\theta,c_-} - s_\theta\big) \tag{6}$$

This means that the attractive field ($\lambda > 0$) directed towards condition $c$ in Eq. (5) is turned into a repulsive field directed away from the condition $c$. Note that conditioning variables and scores associated with an undesired (or negative) condition are denoted in red ($c_-$ and $s_{\theta,c_-}$, respectively). Those referring to wanted (i.e., positive) prompts are written in green ($c_+$ and $s_{\theta,c_+}$). From a likelihood perspective, the NP equation (6) implies sampling from a joint distribution that is inversely proportional to the posterior likelihood $p(\boldsymbol{x}, \boldsymbol{c}) \propto p(\boldsymbol{x})/p(c_-|\boldsymbol{x})^\lambda$.

## 2.3 RECENT WORK RELATED TO NEGATIVE PROMPTING

Despite its adoption in influential models, as described in blogs like (Andrew, 2023; Bhasin, 2024), NP remains understudied on a fundamental level, in particular in terms of how and when it should be applied for optimal effect. For example, Ban et al. (2024) argue that negative prompting is only of practical use once the undesired features have been generated by the positive prompt, whereas Armandpour et al. (2023) propose limiting negative guidance along a direction orthogonal to that of the positive score,to avoid degrading the effect of the positive prompt.
Most similar to our approach are so-called training free guidance schemes (Yu et al., 2023; Du et al., 2023; Brack et al., 2023; Shen et al., 2024). Such approaches often work in a fashion very similar to CG, in which a score field, derived as the gradient of a classifier, is added to the unconditional score field. These do not require any expensive retraining or fine-tuning of the underlying generative model. One method closely related to our work is that proposed by Schramowski et al. (2023), called *Safe Latent Diffusion* (SLD), which use the model's knowledge of certain concepts to avoid generation of undesired features, in a manner very similar to NP. Their scheme differs from NP in two regards. First, they consider a heuristic non-constant guidance scale that is only active when the predictions of the negatively prompted model overlaps with that of the positively prompted model. This approach does not take the explicit time-dependency into account. Second, they consider a pixel wise guidance scale, which allows to *locally* mix the 'allowed' and 'forbidden' outputs of the network. In our work, we do not use this pixel-wise guidance weights as this feature is not present in the analytical formulas with known score functions (see our derivations in Sec. 3.1). Finally, Chen et al. (2024) use a very similar approach to attack the problem of memorization of training samples in DMs. Our approach is compared to that of Schramowski et al. (2023) in more detail in Appendix E.

Most of the interest in negative guidance within the literature is driven by safety considerations. Such an approach can indeed be used to move away from undesired *Not-Safe-For-Work* content (Schramowski et al., 2023). Most approaches on the subject focus on removing entire concepts from diffusion models by fine-tuning certain layers inside the U-Net, whereby typically the attention layers are targeted (Zhang et al., 2024; Gandikota et al., 2023b; Wu et al., 2024; Heng & Soh, 2023; Gandikota et al., 2023a). The most difficult part of fine-tuning approaches is to only remove the targeted concepts without harming the rest (Heng & Soh, 2023). Another option to avoid the generation of certain features in T2I applications is to instead modify the textual prompts themselves before prompting the model (Kumari et al., 2023; Li et al., 2024). A good overview of the different approaches is described by Pham et al. (2024), where the question of how easily such approaches can be bypassed is treated.

## 3 METHODOLOGY

The most obvious problem with NP is that there is a clear risk that $p_t(\boldsymbol{x}, c_-) \propto p_t(\boldsymbol{x})/p_t(c_-|\boldsymbol{x})^\lambda$ is not normalizable. This happens when more than a single point has a zero posterior likelihood, which, especially at low noise levels, is a desired property for an ideal classifier. But even from a score-based perspective, NP is flawed. The score fields are approximately of harmonic nature, implying that the further one is from satisfying the condition $c_+$, the larger the magnitude of $s_{\theta,c_+}$. Simply inverting the score field, as is done in NP, implies the presence of very large score field in regions completely unrelated to $c_-$. This is illustrated and discussed in more detail in Appendix A. What is desired is a guidance field that is strong in regions where $p_t(c_-|\boldsymbol{x}, t) \approx 1$ and very weak where $p_t(c_-|\boldsymbol{x}, t) \approx 0$.

### 3.1 DYNAMIC NEGATIVE GUIDANCE

Instead of simply reverting the sign of the guidance scale, our approach reconstructs the desired score function $s_{\theta,c_+}$ from a linear combination of the unconditional score $s_\theta$ and the unwanted score $s_{\theta,c_-}$. This method should correspond to CG, with instead of the typical posterior conditioned on $c_+$, its complement conditioned on $c_-$. In the case of negative guidance, one has $p_t(c_+|x,t) = 1 - p_t(c_-|x)$. Negative guidance should therefore aim to sample from the following conditional distribution:

$$p_t(x|c_+) \propto p_t(x)\big(1 - p_t(c_-|x)\big) \tag{7}$$

Notice that this joint distribution is always normalizable, as the singularity in $p_t(c_-|x) = 0$ has been removed. From a score based perspective, this joint distribution results in:

$$\nabla_x \log p_t(x|c_+) = \nabla_x \log p_t(x) + \nabla_x \log\big(1 - p_t(c_-|x)\big) \tag{8}$$

Unlike in CFG, the last term is not directly recognizable as a linear combination of scores. Application of the chain rule yields the score function:

$$
\begin{aligned}
\nabla_x \log\big(1 - p_t(c_-|x)\big) &= -\frac{1}{1 - p_t(c_-|x)} \nabla_x p_t(c_-|x) \\
&= -\frac{p_t(c_-|x)}{1 - p_t(c_-|x)} \nabla_x \log p_t(c_-|x)
\end{aligned}
\tag{9}
$$

By then reapplying Bayes's rule on the posterior, and sharpening the posterior by an additional parameter analogous to the guidance scale $\lambda^2$, as done in Ho & Salimans (2021):

$$
\begin{aligned}
\nabla_x \log p_t(x|c_+) &= \nabla_x \log p_t(x) - \lambda_0 \frac{p_t(c_-|x)}{1 - p_t(c_-|x)} \big(\nabla_x \log p_t(x|c_-) - \nabla_x \log p_t(x)\big) \\
&= \nabla_x \log p_t(x) - \lambda(x,t)\big(\nabla_x \log p_t(x|c_-) - \nabla_x \log p_t(x)\big)
\end{aligned}
\tag{10}
$$

This is the equation behind our proposed Dynamic Negative Guidance. It is equivalent to CFG with a hypothetical positive prompt: "everything but $c_-$". The minus sign governing the guidance term, illustrates the repulsive nature of the corresponding guidance field, already present in NP. A key distinction with NP is the presence of the additional $\frac{p_t(c_-|x)}{1-p_t(c_-|x)}$ factor. This is precisely the factor that causes the negative guidance field to be asymptotically large in regions where $p_t(c_-|x) \to 1$ and to be zero in regions where $p_t(c_-|x) \to 0$. Due to its role, similar to the constant guidance scale $\lambda$ in CFG, we decide to reuse the name for our time and state dependent guidance function, i.e. we refer to $\lambda(x,t)$ as the *dynamic* guidance scale. It measures the strength with which the model pushes away from $c_-$ at timestep $t$. Early in the denoising process, the posterior $p(c_-|x)$ corresponds to the prior $p(c_-)$ as large amounts of noise make classification difficult. As the prior is expected to be low, this results in a low guidance scale early on. Only when the classifier becomes confident that $c_-$ is being generated does the negative guidance contribution become more pronounced. When approximating the guidance scale up to zeroth order, a static scheme with constant guidance scale is retrieved, corresponding to NP. It should be noted that such a non-constant guidance scale is in line with recent trends in literature (Kynkäänniemi et al., 2024; Castillo et al., 2023; Schramowski et al., 2023; Wang et al., 2024). Algorithm 1 summarizes the DNG scheme.

To validate our method, the case of diffusion of one dimensional Gaussian mixtures is considered. In this setting, the likelihoods as well as the posterior are analytically tractable throughout the diffusion process $t$. The goal is to guide away from a single of the Gaussian modes. The mathematical details are given in appendix B. These experiments, shown in Fig. 1, reveal that using DNG conditioned on the unwanted mode with the exact posterior is equivalent to using CFG conditioned on the desired modes. This is in stark contrast to what is obtained when using NP (visible in Fig. 1c), which completely fails to model the target distribution, demonstrating the theoretical shortcomings of the negative prompting algorithm. It should be noted, that while this failure is very apparent in one dimension, it does not necessarily imply that such a strong statement can be made in arbitrary high dimensional spaces.

---

[2]This implies sampling from $p_t(x|c_-) \propto p_t(x)\big(1 - p_t(c_-|x)\big)^\lambda$

---

**Algorithm 1** Dynamic Negative Guidance

---

**Input:** Pre-trained unconditional DDPM with noise prediction $\boldsymbol{\epsilon}_\theta$, Pre-trained *to-forget* DDPM with noise prediction $\boldsymbol{\epsilon}_{f,\theta}$, guidance scale $\lambda_0$, prior $p_0$ and Temperature $\tau$

$\boldsymbol{z} \sim \mathcal{N}(0, \boldsymbol{I}), \boldsymbol{x}_T \sim \mathcal{N}(0, \boldsymbol{I})$

$p(\boldsymbol{c}\text{-}|\boldsymbol{x}_T) = p_0$                          `Initialize posterior and guidance scale`

$\boldsymbol{\lambda_T}(\boldsymbol{x}_T) = \lambda_0 \frac{p(\boldsymbol{c}\text{-}|\boldsymbol{x}_T)}{1-p(\boldsymbol{c}\text{-}|\boldsymbol{x}_T)}$

**for** $t = T, \dots, 1$ **do**

    $\boldsymbol{\epsilon}_{\theta,\text{guid}}(\boldsymbol{x}_t) = \boldsymbol{\epsilon}_\theta(\boldsymbol{x}_t) - \lambda_t(\boldsymbol{x}_t)\big(\boldsymbol{\epsilon}_{f,\theta}(\boldsymbol{x}_t) - \boldsymbol{\epsilon}_\theta(\boldsymbol{x}_t)\big)$             `Apply guidance`

    $\boldsymbol{x}_{t-1} = \frac{1}{\sqrt{\alpha_t}}\big(\boldsymbol{x}_t - \frac{1-\alpha_t}{\sqrt{1-\bar{\alpha}_t}}\boldsymbol{\epsilon}_{\theta,\text{guid}}(\boldsymbol{x}_t)\big) + \sqrt{\beta_t}\boldsymbol{z}$                `DDPM Step`

    $p(\boldsymbol{c}\text{-}|\boldsymbol{x}_{t-1}) = \text{Compute posterior}\big(p(\boldsymbol{c}\text{-}|\boldsymbol{x}_t), \boldsymbol{x}_{t-1}, \boldsymbol{x}_t, \boldsymbol{\epsilon}_\theta(\boldsymbol{x}_t), \boldsymbol{\epsilon}_{f,\theta}(\boldsymbol{x}_t)\big)$    `See Algorithm 2`

    $\lambda_t(\boldsymbol{x}_{t-1}) = \lambda_0 \frac{p(\boldsymbol{c}\text{-}|\boldsymbol{x}_{t-1})}{1-p(\boldsymbol{c}\text{-}|\boldsymbol{x}_{t-1})}$                       `Compute new guidance scale`

**end for**

---

## 3.2 POSTERIOR APPROXIMATION THROUGH MARKOV CHAIN

Our novel *dynamic* guidance scale $\lambda(\boldsymbol{x}, t)$ relies on the posterior likelihood $p_t(\boldsymbol{c}\text{-}|\boldsymbol{x})$, which is in general not available using score-based models. Inspired by ideas from Li et al. (2023), we propose to approximate the required posterior $p(\boldsymbol{c}\text{-}|\boldsymbol{x}, t)$ by estimating the required likelihoods by tracking the diffusion Markov Chain, defined by Eq. (1) during the denoising process. All the infinitesimal transition probabilities are approximately Gaussian $p_s(\boldsymbol{x}_s|\boldsymbol{x}_{s+1}; \theta) \approx \mathcal{N}(\boldsymbol{x}_s; \boldsymbol{\mu}_{s,\theta}(\boldsymbol{x}_{s+1}), \sigma_s^2 \boldsymbol{I})$ where $\boldsymbol{\mu}_{s,\theta}(\boldsymbol{x}_{s+1})$ can be retrieved from the noise predictions $\boldsymbol{\epsilon}_{s,\theta}(\boldsymbol{x}_{s+1})$ using Eq. (2). To compute the dynamic guidance scale specified by Eq. (10) the posterior can be estimated from the likelihoods using Bayes rule:

$$p_t(\boldsymbol{c}\text{-}|\boldsymbol{x}_{t:T}) = p(\boldsymbol{c}\text{-})\frac{p(\boldsymbol{x}_{t:T}|\boldsymbol{c}\text{-})}{p(\boldsymbol{x}_{t:T})}$$

$$\iff \log p_t(\boldsymbol{c}\text{-}|\boldsymbol{x}_{t:T}) = \log p(\boldsymbol{c}\text{-}) + \sum_{i=t}^{T-1}\big(\log p_i(\boldsymbol{x}_i|\boldsymbol{x}_{i+1}, \boldsymbol{c}\text{-}; \theta) - \log p_i(\boldsymbol{x}_i|\boldsymbol{x}_{i+1}; \theta)\big) \quad (11)$$

$$= \log p_{t+1}(\boldsymbol{c}\text{-}|\boldsymbol{x}_{t+1:T}) + \log p_t(\boldsymbol{x}_t|\boldsymbol{x}_{t+1}, \boldsymbol{c}\text{-}; \theta) - \log p_t(\boldsymbol{x}_t|\boldsymbol{x}_{t+1}; \theta)$$

It should be noted that the posterior $p_t(\boldsymbol{c}\text{-}|\boldsymbol{x}_{t:T})$, is independent of the diffusion trajectory $\boldsymbol{x}_{t:T}$ and only depends on the least noisy time step $p_t(\boldsymbol{c}\text{-}|\boldsymbol{x}_{t:T}) = p_t(\boldsymbol{c}\text{-}|\boldsymbol{x}_t)$. This is a consequence of the Markov Assumption and is proven in Appendix C. The transition probability of the negatively guided model is also Gaussian of equal variance, i.e. $p_t(\boldsymbol{x}_t|\boldsymbol{x}_{t+1}, \boldsymbol{c}\text{-}; \theta) \approx \mathcal{N}(\boldsymbol{x}_t; \boldsymbol{\mu}_{t,\theta}(\boldsymbol{x}_{t+1}|\boldsymbol{c}\text{-}), \sigma_t^2 \boldsymbol{I})$. We therefore obtain the update rule:

$$\log p_t(\boldsymbol{c}\text{-}|\boldsymbol{x}_t) = \log p_{t+1}(\boldsymbol{c}\text{-}|\boldsymbol{x}_{t+1}) - \frac{1}{2\sigma_t^2}\big(\|\boldsymbol{x}_t - \boldsymbol{\mu}_{t,\theta}(\boldsymbol{x}_{t+1}|\boldsymbol{c}\text{-})\|^2 - \|\boldsymbol{x}_t - \boldsymbol{\mu}_{t,\theta}(\boldsymbol{x}_{t+1})\|^2\big) \quad (12)$$

To know in which point $\boldsymbol{x}_t$ the posterior needs to be estimated, the guidance scale is required, which depends itself on the posterior. An implicit problem is therefore defined. To resolve the implicitness of the above equation, we assume that the posterior changes slowly such that the guidance scale can, up to first order, be approximated by its previous value. This assumption becomes exact as the number of diffusion time steps becomes infinitely large, an assumption often used in the diffusion literature. In essence, the computation of the guidance scale and that of the denoising is staggered in time. To obtain the guidance scale required to find $\boldsymbol{x}_{t-1}$, the posterior at time step $t$ is used, which solely depends on $\boldsymbol{x}_t$, $\boldsymbol{\mu}_{t,\theta}(\boldsymbol{x}_{t+1})$ and $\boldsymbol{\mu}_{t,\theta}(\boldsymbol{x}_{t+1}|\boldsymbol{c}\text{-})$, which are all known. Mathematically, this boils down to replacing $\log p_t(\boldsymbol{c}\text{-}|\boldsymbol{x}_t)$ with $\log p_{t-1}(\boldsymbol{c}\text{-}|\boldsymbol{x}_{t-1})$ in Eq. (12).

The term added to the posterior in Eq. (12) can be positive or negative, respectively corresponding to an increase or a decrease of the posterior likelihood. When the new state is closer to the unwanted mean $\boldsymbol{\mu}_{t,\theta}(\boldsymbol{x}_{t+1}|\boldsymbol{c}\text{-})$ than to the unconditional one $\boldsymbol{\mu}_{t,\theta}(\boldsymbol{x}_{t+1})$, the posterior, and hence the guidance scale $\lambda(\boldsymbol{x}, t)$, increases. It should be noted that while the unwanted mean prediction $\boldsymbol{\mu}_{t,\theta}(\boldsymbol{x}_{t+1}|\boldsymbol{c}\text{-})$ can sometimes be totally off (imagine a model that can only generate cats that is trying to denoise the image of a truck), the unconditional mean $\boldsymbol{\mu}_{t,\theta}(\boldsymbol{x}_{t+1})$ is always a good denoiser (using the previous example, the unconditional model can generate both cats and trucks and therefore has no problem denoising any of the two). This has as consequence that the likelihood decreasing terms are much larger in magnitude than the likelihood increasing ones. This is explained visually in Fig. 2.

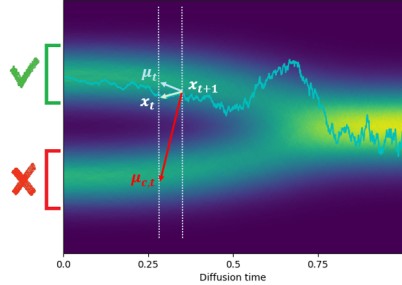

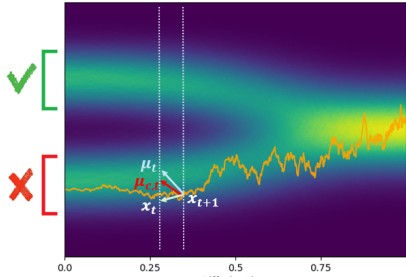

(a) When generating allowed feature

(b) When generating forbidden feature

Figure 2: Visualization of the different components for estimating the posterior. In (a) an allowed point is being generated, in which circumstances $\|\boldsymbol{x}_t - \boldsymbol{\mu}_t\| \ll \|\boldsymbol{x}_t - \boldsymbol{\mu}_{c,t}\|$, corresponding to a large decrease of the posterior. In (b) a forbidden point is being generated, in which circumstances it is possible that $\|\boldsymbol{x}_t - \boldsymbol{\mu}_t\| > \|\boldsymbol{x}_t - \boldsymbol{\mu}_{c,t}\|$ corresponding to a slight increase of the posterior probability.

To regularize this effect, we propose adding a linear transformation to the difference of Euclidean distances. Rescaling the difference by a factor $\tau \in ]0, 1[$ can diminish stochastic fluctuations present during denoising, while a small offset $\delta$ creates a slight bias towards increasing the posterior. Should an allowed image be generated, this offset is completely dominated by the very large difference. But should the predictions of the unconditional and the negatively conditioned model be similar, the posterior would increase. The estimation of the posterior probability is summarized in Algorithm 2.

---

**Algorithm 2** Compute posterior

---

**Input:** Previous estimation of filtering posterior $p_t(c\text{-}|\boldsymbol{x}_t)$, Updated noisy state $\boldsymbol{x}_{t-1}$, previous noisy state $\boldsymbol{x}_t$, unconditional noise prediction $\boldsymbol{\epsilon}_\theta(\boldsymbol{x}_t)$, *to-forget* noise prediction $\boldsymbol{\epsilon}_{f,\theta}(\boldsymbol{x}_t)$, diffusion constants $\alpha_t$, $\bar{\alpha}_t$, prior $p_0$, Temperature $\tau$, offset $\delta$, minimal and maximal posterior values $p_{\min}$ and $p_{\max}$

$\sigma_t^2 = 1 - \alpha_t$                    Variance of Gaussian at $t$

$\boldsymbol{\mu}(\boldsymbol{x}_t) = \frac{1}{\sqrt{\alpha_t}} \left( \boldsymbol{x}_t - \frac{1-\alpha_t}{\sqrt{1-\bar{\alpha}_t}} \boldsymbol{\epsilon}_{f,\theta}(\boldsymbol{x}_t) \right)$                    *Undesired* mean prediction

$\boldsymbol{\mu}(\boldsymbol{x}_t) = \frac{1}{\sqrt{\alpha_t}} \left( \boldsymbol{x}_t - \frac{1-\alpha_t}{\sqrt{1-\bar{\alpha}_t}} \boldsymbol{\epsilon}_\theta(\boldsymbol{x}_t) \right)$                    Unconditional mean prediction

$p(c\text{-}|\boldsymbol{x}_{t-1}) = p_t(c\text{-}|\boldsymbol{x}_t) \exp \left( -\frac{\tau}{2\sigma_t^2} \left( \|\boldsymbol{x}_{t-1} - \boldsymbol{\mu}(\boldsymbol{x}_t)\|^2 - \|\boldsymbol{x}_{t-1} - \boldsymbol{\mu}(\boldsymbol{x}_t)\|^2 \right) + \frac{\delta}{2\sigma_t^2} \right)$

$p(c\text{-}|\boldsymbol{x}_{t-1}) = \text{Clamp}\left( p(c\text{-}|\boldsymbol{x}_{t-1}), \min = p_{\min}, \max = p_{\max} \right)$

**Output:** Approximate posterior probability $p(c\text{-}|\boldsymbol{x}_{t-1})$

---

To illustrate the validity of the proposed posterior approximation scheme, the one dimensional Gaussian case can again be analyzed. The approximated posterior can be compared to the known exact posterior. To remove any effects possibly caused by the guidance itself, the posterior is tracked at zero-guidance scale ($\lambda = 0$). To visualize the results, 100k points are sampled using the unconditional model and then classified into two groups based on whether they belong to the forbidden mode[3]. For both groups, the exact and approximated posterior values are averaged at each diffusion time step, as shown in Fig. 3a, demonstrating the validity of the approximation. An example of a distribution sampled using our dynamic negative guidance with the approximate posterior is shown in Fig. 3b. The derived approach is highly flexible, solely requiring a conditional diffusion model. It remains valid using latent representations, such as used in modern Latent Diffusion Models (Rombach et al., 2022).

## 4 EXPERIMENTAL RESULTS

### 4.1 CLASS REMOVAL

The proposed algorithm is tested in the context of image generation on labelled datasets, in the present case MNIST and CIFAR10 are considered. The task is to remove one of the classes by guiding an

---

[3]The forbidden mode is chosen to be distinct from others to simplify the classification task (with the unconditional distribution shown in red in Fig. 3b)

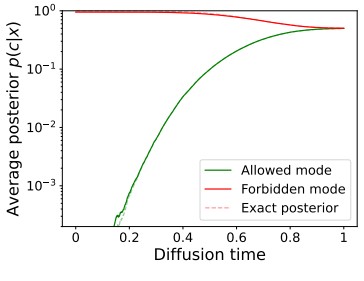

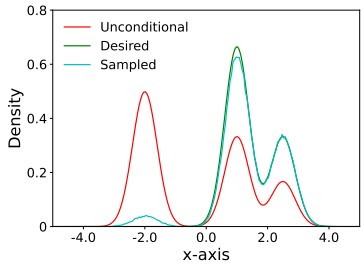

(a) Posterior estimation ($\lambda = 0$)

(b) Sampled distribution using approx. posterior ($\lambda = 1$)

Figure 3: Analysis of the posterior estimation. In (a) the dotted lines represent the exact posterior, while the full lines represent the approximated posterior. The color indicates whether the generated samples belong to the forbidden region or not. These samples were generated without guidance. (b) an example of a distribution sampled using the approximate posterior with guidance scale $\lambda = 1$.

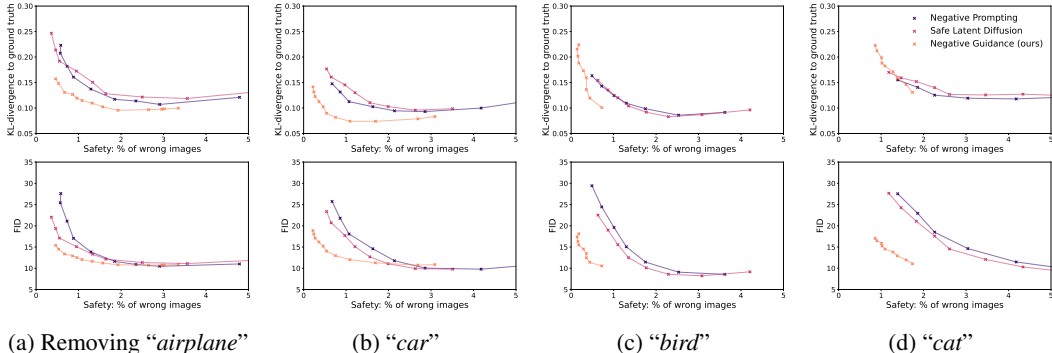

(a) Removing "*airplane*"  (b) "*car*"  (c) "*bird*"  (d) "*cat*"

Figure 4: Evaluation of the KL-divergence (top) and FID (bottom) on single class removal experiments performed on CIFAR10 using Negative Prompting, Safe Latent Diffusion and our Dynamic Negative Guidance. Despite differences between classes, DNG outperforms both NP and SLD in all settings.

unconditional model with a model trained solely on that specific class. To quantify the *safety* of the approach, a classifier assesses the percentage of generated images that belong to the forbidden class, while the *diversity* is measured by examining the overall distribution of generated classes across all images. Ideally, this distribution should contain a single zero and equal weight on all other classes (see Figure 13). To measure how well this ideal case is approximated, the KL-divergence between ideal and generated distributions is computed. To observe how the *quality* of the model is altered, the standard FID metric (Heusel et al., 2017) is used. Being a statistical tool, the FID is also affected by large class imbalances. The FID not only measures the quality of the images but also accounts for the diversity of the model. It still remains the most widespread quality metric for image generation and has time and again been proven to coincide with human perception. To compute the FID the statistics of 10240 generated images is compared to the training data without the undesired class.

To compare different approaches, both the FID and the KL-divergence are compared with the safety of the method when sweeping over the initial guidance scale $\lambda$. These graphs are shown when removing various classes from CIFAR10 in Fig. 4. Similar results are obtained on MNIST. These are shown in Fig. 14 in appendix G. Especially relevant is the regime of high safety, where as few as possible forbidden images are generated. In this regime, DNG outperforms both NP and SLD on all four classes, showcasing that the image removal performed by DNG is less invasive than previous approaches. Different values of the hyperparameters of SLD are tested and reported in appendix E, our approach is compared with the setting which performs best at high safety levels.

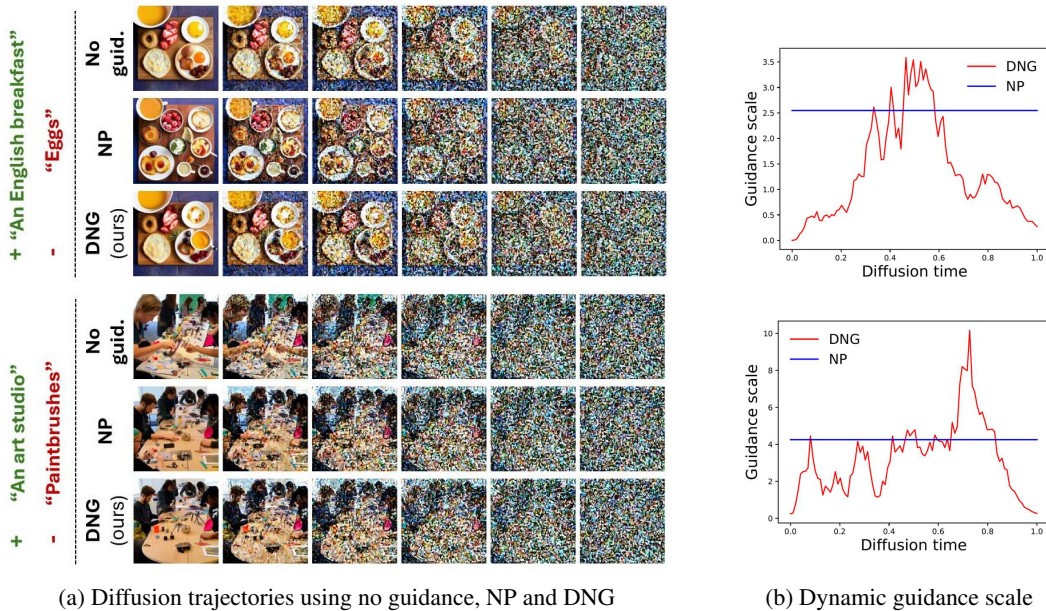

(a) Diffusion trajectories using no guidance, NP and DNG        (b) Dynamic guidance scale

Figure 5: In 5a, illustrative diffusion trajectories using Stable Diffusion on different prompts are shown. Notice that DNG only removes the unwanted object and keeps, for instance, the bread in the bottom left of the top example. In 5b, the dynamic guidance scale for the specific trajectories is compared to that used for NP. At which timestep the guidance is most active depends on the specific diffusion trajectory as well as the prompt.

## 4.2 TEXT-TO-IMAGE

To demonstrate the flexibility of the proposed DNG scheme, small scale experiments are run using Stable Diffusion 2 (Rombach et al., 2022). Inspired by Schramowski et al. (2023), the role of the unguided model is replaced by that of the positively prompted one. Mathematically, $s_\theta$ is substituted by $s_{\theta,c_+}$ in Algorithm 2. To compare the different negative guidance schemes in the context of T2I, prompts that implicitly generate objects are required. An example could be an "English breakfast", for which it is highly likely that the model generates an image that contains an egg. The term "egg" can then be chosen as negative prompt. Using ChatGPT we design 5 prompts that are highly likely to generate certain objects without these features being explicitly mentioned in the prompt. These 5 prompts are given in appendix D. To illustrate the time dependence of our approach, diffusion trajectories are shown for NP and DNG as well as the unguided case (i.e., without any negative guidance) in Fig. 5a. The dynamic guidance scale computed through our posterior estimation is compared to the constant one used in NP in Fig. 5b. We observe that our dynamic guidance scale is most active at different timesteps depending both on the prompts and the diffusion trajectory itself. In agreement with current literature we find that the sooner it is activated, the larger the changes in the resulting image are Kynkäänniemi et al. (2024); Wang et al. (2024). To evaluate how well DNG preserves the diversity of the underlying model, we examine how the guidance deactivates itself when the undesired element is not present. To achieve this, we generate a second image with a negative prompt consisting of a semantically unrelated concept. For example, with "English breakfast", an unrelated concept could be "The view of a skyline". In that case, DNG should let the posterior $p(c_-|x, t)$, and hence the guidance scale, go to zero. A schematic representation of this analysis is shown in Fig. 6.It shows that while both DNG and NP are able to remove the related negative prompt ("*an egg*") from the original image, only NP modifies the image when using an unrelated negative prompt (for instance the tea cup on the top left of the image). We acknowledge that the images generated using NP with an unrelated negative prompt still follow the positive prompt. The fact they are altered unnecessarily implies a potential loss of image diversity Ho & Salimans (2021); Wang et al. (2024); Kynkäänniemi et al. (2024). To quantify how much the guided images have been altered by the guidance scheme, the CLIP-score, i.e. the cosine similarity of their respective CLIP scace representations, is measured. The shown scores are averaged over 32 images generated using

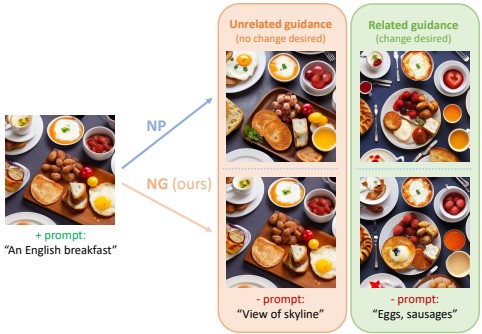 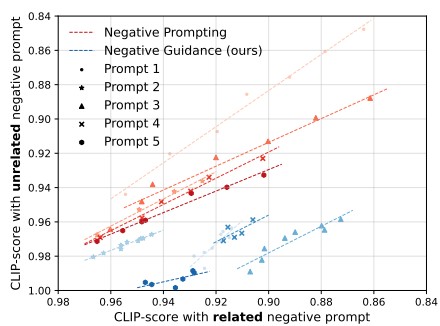

(a) Schematic of the evaluation approach on Stable Diffusion.

(b) CLIP scores in related and unrelated cases

Figure 6: Comparison of NP and DNG on Stable Diffusion. In 6a the evaluation strategy is explained schematically. A fixed seed is denoised using the same positive prompts with two separate negative prompts, a semantically related one and a semantically unrelated one. This is done for DNG and NP. The CLIP score between the guided and unguided images is then measured for both related and unrelated cases and displayed as a function of each other for different prompts in 6b

both NP and DNG for various guidance strengths. Ideally, the CLIP score should remain equal to one when negatively prompted with an unrelated text, indicating that the negative guidance was fully deactivated. On the contrary, the model prompted with a related text should display a decrease of CLIP score as a function of increasing guidance scale. To compare our results to those obtained with NP, the average CLIP score of the samples guided with an unrelated negative prompt can be plotted with respect to the average CLIP score of the samples guided with a related negative prompt. For intuitiveness, the axes are scaled from 1 to -1, such that points located farther right and higher on the graph correspond to greater deviation from the original images, indicated by a lower CLIP score. This is visible in Fig. 6b in which each prompt is given its own marker and color intensity. As both DNG and NP follow linear relationships per prompt, they are approximated by a linear regression in a limited interval. As seen in Figure 6b, our method remains consistent with predictions, causing minimal changes when a semantically unrelated negative prompt is used, while still performing on par with NP in the case of a related negative prompt. Generated samples are provided in Appendix H.

## 5 LIMITATIONS

The main limitation of the present work is the non-exhaustive analysis of our approach in the context of T2I generation. The current results obtained with Stable Diffusion solely demonstrate that the proposed approach shows promising potential. A more thorough analysis using a larger prompt dataset, advanced metrics such as the FID, as well as more generic hyperparameter values, are all still required. Another limitation of the Stable Diffusion experiments, is that no metric is used to verify how well the negative prompt has been removed from the image. This should be done using a visual object classifier. All these steps are left as future work.

## 6 CONCLUSION

Understanding the theoretical flaws behind the widespread Negative Prompting algorithm, led to a novel, theoretically grounded Dynamic Negative Guidance scheme. This scheme was first validated in a low dimensional setting, before being compared to alternative approaches in image generation tasks. In particular, the proposed scheme outperformed concurrent approaches, such as Negative Prompting or Safe Latent Diffusion, at the task of class removal from an unconditional MNIST or CIFAR10 model. The advantages of the dynamic nature of the scheme was further displayed in the context of T2I generation using Stable Diffusion, in which it was shown that DNG deactivates itself when the negative prompt is unrelated to the positive prompt.

ACKNOWLEDGMENTS

This research was partly funded by the Research Foundation - Flanders (FWO-Vlaanderen) under grant G0C2723N and by the Flemish Government (AI Research Program). Gabriel Raya was funded by the Dutch Research Council (NWO) as part of the CERTIF-AI project (file number 17998).

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

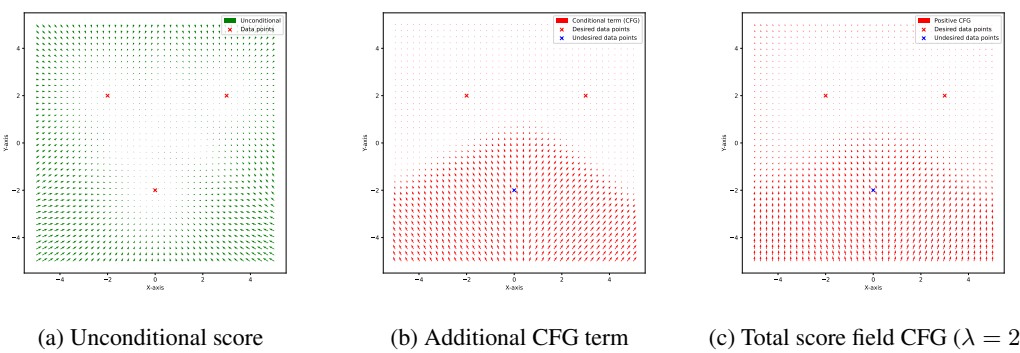

(a) Unconditional score      (b) Additional CFG term      (c) Total score field CFG ($\lambda = 2$)

Figure 7: Illustration of CFG in 2D

## A    ILLUSTRATION OF THE LIMITATIONS OF NEGATIVE PROMPTING

To understand the working principles behind different guidance schemes, it is helpful to analyze the score fields of Gaussian mixtures in two dimensions. Even if conclusions taken in low dimensional spaces do not necessarily transfer to higher dimensions, these can provide valuable insights.

The choice of Gaussian mixtures is not an arbitrary one. Such a score field is the same as that generated by a diffusion model in the memorization regime, with as data points the means of the Gaussian modes (Ho et al., 2020). In that case the network behaves as an associative memory system (Ambrogioni, 2024b; Hoover et al., 2023). To be specific, Gaussian mixtures represent the analytical diffusion of delta peaks centered around specific data points, which can be thought of as memorized training samples. To illustrate the flaws of the NP algorithm, the individual components of the total *guided* score field are plotted. The most relevant term being the so-called guidance term, i.e. the term that is added on top of the unconditional score.

In the example that is discussed, the three memorized data points are shown using red crosses (Fig. 7(a)). The goal is to guide the model towards the two upper modes of the mixture, or, inversely, away from the bottom mode.

For CFG the total field is described by the following equation:

$$s_{CFG} = s_{\boldsymbol{\theta}} + \lambda\big(s_{\boldsymbol{\theta},c_+} - s_{\boldsymbol{\theta}}\big) \tag{13}$$

In this case $s_{\boldsymbol{\theta},c_+}$ can be interpreted as the score field computed using as distribution the Gaussian mixture consisting solely of the two desired modes. To understand the shape of this additional classifier term (i.e. $s_{\boldsymbol{\theta},c_+} - s_{\boldsymbol{\theta}}$), it is useful to think about the problem through the lens of CG. This additional term represents the gradient of log likelihood of the posterior $p(c_+|\boldsymbol{x}, t)$. This likelihood can be computed analytically by using Bayes rule $p(c_+|\boldsymbol{x}, t) = p(c_+)\frac{p(\boldsymbol{x}|c_+, t)}{p_t(\boldsymbol{x})}$. It is constant and equal to one close to the desired points, while it changes when approaching the undesired point as at that point $p_t(\boldsymbol{x}) \ll p(c_+)p(\boldsymbol{x}|c_+, t)^4$. Hence, the additional term is small surrounding the desired points, while it is large close to the undesired point. The direction of the vector field, is specified by the fact that the posterior likelihood decreases when moving closer to the undesired point, implying an attractive field. All these conclusions are clearly visible in Fig. 7.

On the other hand, when using NP, the conditional field is specified by the *undesired* data point $c_-$. The total field is then described by the following equation, with $\lambda$ a positive constant:

$$s_{NP} = s_{\boldsymbol{\theta}} - \lambda\big(s_{\boldsymbol{\theta},c_-} - s_{\boldsymbol{\theta}}\big) \tag{14}$$

The easiest way to understand this score field is to first understand how $\nabla_{\boldsymbol{x}} \log p(c_-|\boldsymbol{x}, t) = s_{\boldsymbol{\theta},c_-} - s_{\boldsymbol{\theta}}$ behaves, and then simply reverse its direction. Essentially this boils down to the case treated above for CFG, with as only difference that desired and undesired modes have swapped position. Close to the undesired mode, the posterior is constant and equal to one, corresponding with a very weak score field. On the other hand, as one approaches the desired means, the posterior drops, implying a

---

[4]This change is perpetuated when $p(c_+|\boldsymbol{x}, t) \to 0$ due to the singularity of the logarithm function in zero, causing an increasing gradient the more the posterior likelihood decreases.

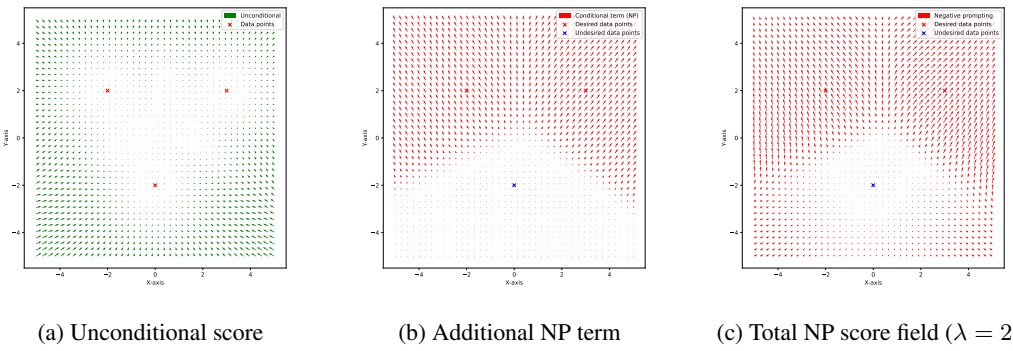

(a) Unconditional score      (b) Additional NP term      (c) Total NP score field ($\lambda = 2$)

Figure 8: Illustration of NP in 2D

large repulsive field from the desired modes. Due to the additional minus sign, this repulsive field of the desired modes becomes attractive, explaining why NP is sensible. The issue is naturally that the field is much stronger in regions that are far away from the undesired mode. What is really desired is, as present for CFG, a strong repulsive guidance field surrounding the undesired mode and a close to zero field as one approaches the desired modes. This is an intuitive explanation of the reasoning behind our proposed DNG scheme. Mathematically, we have shown in section 3.1 that using DNG with $c_-$ is equivalent to using CFG with $c_+ = \bar{c}_-$. It is therefore no surprise that Fig. 9 is equivalent to Fig. 7. The total field described by DNG can be decomposed as:

$$\boldsymbol{s}_{NG} = \boldsymbol{s}_{\boldsymbol{\theta}} - \lambda \frac{p(\boldsymbol{c}_-|\boldsymbol{x})}{1 - p(\boldsymbol{c}_-|\boldsymbol{x})} \left( \boldsymbol{s}_{\boldsymbol{\theta},\boldsymbol{c}_-} - \boldsymbol{s}_{\boldsymbol{\theta}} \right) \tag{15}$$

The difference with NP is the presence of a state dependent guidance scale. This guidance scale is proportional to $p(\boldsymbol{c}_-|\boldsymbol{x})$, which causes the guidance term to drop to zero when $\boldsymbol{x}$ is far from the undesired point. Additionally, the guidance scale becomes asymptotically large as $\boldsymbol{x}$ approaches the undesired point (i.e. when $p(\boldsymbol{c}_-|\boldsymbol{x}, t) \to 1$). The guidance score field now posses both desired properties, it is not only directed away from the undesired point, but crucially it is also larger in regions closer to that point. The results described here can leave one wondering why NP even works in practice. It should however not be forgotten that the picture here is not only oversimplified by the choice of low dimensionality of the problem but also by the choice of very simple, well separated Gaussian modes. In reality the distribution of natural images is much more complex such that the posterior $p(\boldsymbol{c}_-|\boldsymbol{x}, t)$ follows a much more complex pattern. While this clearly illustrates the fundamental shortcomings of NP, it is important not to over-interpret these results. NP has proven to be a valuable tool for DMs, and more research is required to better understand how its theoretical shortcomings impact it at large scale.

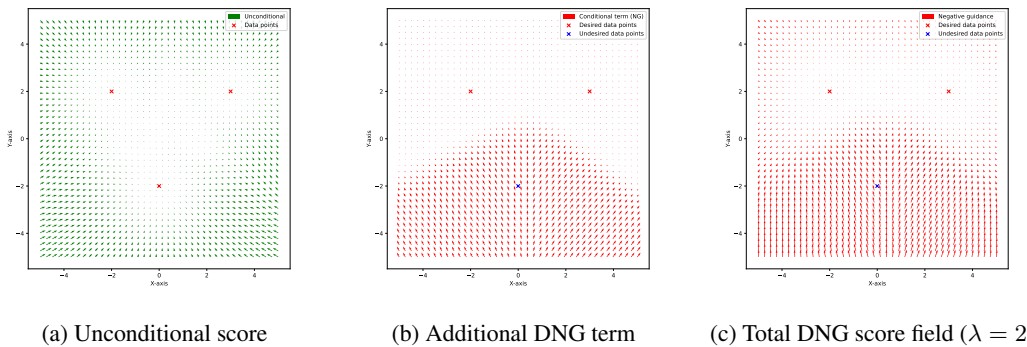

(a) Unconditional score      (b) Additional DNG term      (c) Total DNG score field ($\lambda = 2$)

Figure 9: Illustration of DNG in 2D

## B  ONE-DIMENSIONAL DIFFUSION EXAMPLE: DETAILS

To validate the theory described in the main body, a one-dimensional diffusion process can be analyzed. For this purpose, it is useful to choose the sample distribution $p(\boldsymbol{x}, 0)$ as a Gaussian mixture. One of the advantage of this choice is its good fit with the typically used Gaussian diffusion kernel, making everything easily tractable.

Specifically, starting from an unperturbed distribution defined by $p(\boldsymbol{x}, 0) = \sum_{i=1}^{N} c_i \mathcal{N}\big(\boldsymbol{x}; \boldsymbol{\mu_i}, \sigma_i^2 \boldsymbol{I}\big)$, it is straightforward to show that the discrete variance preserving diffusion process at time step $t$ results in the following time dependence $p_t(\boldsymbol{x}) = \sum_{i=1}^{N} c_i \mathcal{N}\big(\boldsymbol{x}; \sqrt{\bar{\alpha}_t} \boldsymbol{\mu_i}, (1 - \bar{\alpha}_t + \bar{\alpha}_t \sigma_i^2) \boldsymbol{I}\big)$. By choosing a Gaussian mixture as underlying distribution, the diffused distribution remains a sum of Gaussians at every time step. In practice, this distribution is modeled by its score, denoted by $s(\boldsymbol{x}, t)$. To validate our theory, some of the modes, say $M < N$, are removed at inference through different guidance schemes. For this purpose, two helpful distributions are defined, the *to-forget* distribution $p_f(\boldsymbol{x}, 0) = \sum_{i=1}^{N_f} c_i' \mathcal{N}\big(\boldsymbol{x}; \boldsymbol{\mu_i}, \sigma_i^2 \boldsymbol{I}\big)$ and the *to-remember* distribution $p_r(\boldsymbol{x}, 0) = \sum_{i=N_f+1}^{N} c_i'' \mathcal{N}\big(\boldsymbol{x}; \boldsymbol{\mu_i}, \sigma_i^2 \boldsymbol{I}\big)$ [5]. Both of these distributions are implicitly defined through their respective score functions $s_f(\boldsymbol{x}, t)$ and $s_r(\boldsymbol{x}, t)$. The goal of negative guidance can be resumed as trying to sample $p_r(\boldsymbol{x}, 0)$ by using a combination of the unconditional score $s(\boldsymbol{x}, t)$ and the *to-forget* score $s_f(\boldsymbol{x}, t)$.

Another advantage of working in this one dimensional setting is that the posterior is analytically available through Bayes rule:

$$p(c = 0 | \boldsymbol{x}, t) = p(c = 0) \frac{p(\boldsymbol{x}, t | c = 0)}{p_t(\boldsymbol{x})} = p(c = 0) \frac{p_f(\boldsymbol{x}, t)}{p_t(\boldsymbol{x})} \tag{16}$$

The theory can therefore easily be verified using analytical solutions. Negative guidance, with the exact posterior using $s(\boldsymbol{x}, t)$ and $s_f(\boldsymbol{x}, t)$, should exactly correspond to positive guidance using $s(\boldsymbol{x}, t)$ and $s_r(\boldsymbol{x}, t)$. In practice, we demonstrate that:

$$\text{Positive guidance}\big(s(\boldsymbol{x}, t), s_r(\boldsymbol{x}, t)\big) = \text{Negative guidance}\big(s(\boldsymbol{x}, t), s_f(\boldsymbol{x}, t), p(c = 0 | \boldsymbol{x}, t)\big)$$

$$s(\boldsymbol{x}, t) + \lambda\big(s_r(\boldsymbol{x}, t) - s(\boldsymbol{x}, t)\big) = s(\boldsymbol{x}, t) - \lambda \frac{p(c = 0 | \boldsymbol{x}, t)}{1 - p(c = 0 | \boldsymbol{x}, t)} \big(s_f(\boldsymbol{x}, t) - s(\boldsymbol{x}, t)\big) \tag{17}$$

## C  INDEPENDENCE OF POSTERIOR DISTRIBUTION AND DIFFUSION TRAJECTORY

The likelihood of the entire diffusion path can be decomposed through Bayes rule:

$$\begin{aligned} p(\boldsymbol{x}_{t:T}) &= p(\boldsymbol{x}_t, \boldsymbol{x}_{t+1}, \cdots, \boldsymbol{x}_T) \\ &= p(\boldsymbol{x}_{t+1}, \cdots, \boldsymbol{x}_T | \boldsymbol{x}_t) p(\boldsymbol{x}_t) \\ &= p(\boldsymbol{x}_t) p(\boldsymbol{x}_{t+1:T} | \boldsymbol{x}_t) \end{aligned} \tag{18}$$

Doing this for both the conditional model $p(\cdot | c = 0)$ and the unconditional model $p(\cdot)$ and inserting this in the equation of the posterior (i.e. Eq (11)):

$$\begin{aligned} p(c = 0 | \boldsymbol{x}_{t:T}) &= p(c = 0) \frac{p(\boldsymbol{x}_{t:T} | c = 0)}{p(\boldsymbol{x}_{t:T})} \\ &= p(c = 0) \frac{p(\boldsymbol{x}_t | c = 0) p(\boldsymbol{x}_{t+1:T} | \boldsymbol{x}_t, c = 0)}{p(\boldsymbol{x}_t) p(\boldsymbol{x}_{t+1:T} | \boldsymbol{x}_t)} \\ &= p(c = 0) \frac{p(\boldsymbol{x}_t | c = 0)}{p(\boldsymbol{x}_t)} \frac{p(\boldsymbol{x}_{t+1:T} | \boldsymbol{x}_t, c = 0)}{p(\boldsymbol{x}_{t+1:T} | \boldsymbol{x}_t)} \end{aligned} \tag{19}$$

The term $p(\boldsymbol{x}_{t+1:T} | \boldsymbol{x}_t)$ refers to the forward process, which corresponds to the fixed, known forward process denoted by $q(\boldsymbol{x}_{t+1:T} | \boldsymbol{x}_t)$, i.e. one has $p(\boldsymbol{x}_{t+1:T} | \boldsymbol{x}_t) \simeq q(\boldsymbol{x}_{t+1:T} | \boldsymbol{x}_t)$. Both the conditional and unconditional model follow the same forward process such that the last term of Eq.(19) drops

---

[5]The weighting constants for $p_f(\boldsymbol{x}, 0)$ and $p_r(\boldsymbol{x}, 0)$ are renormalised, i.e. $c_i', c_i'' \propto c_i$ with $\sum_{i=0}^{M} c_i' = 1, \sum_{i=M+1}^{N} c_i'' = 1$

out, i.e. $\frac{p(\boldsymbol{x}_{t+1:T}|\boldsymbol{x}_t,c=0)}{p(\boldsymbol{x}_{t+1:T}|\boldsymbol{x}_t)} \simeq \frac{q(\boldsymbol{x}_{t+1:T}|\boldsymbol{x}_t)}{q(\boldsymbol{x}_{t+1:T}|\boldsymbol{x}_t)} = 1$.

Using this, it is easy to show that the posterior is indeed independent of the trajectory taken to reach $\boldsymbol{x}_t$:

$$
\begin{aligned}
p(c=0|\boldsymbol{x}_{t:T}) &= p(c=0)\frac{p(\boldsymbol{x}_t|c=0)}{p(\boldsymbol{x}_t)}\frac{p(\boldsymbol{x}_{t+1:T}|\boldsymbol{x}_t,c=0)}{p(\boldsymbol{x}_{t+1:T}|\boldsymbol{x}_t)} \\
&\simeq p(c=0)\frac{p(\boldsymbol{x}_t|c=0)}{p(\boldsymbol{x}_t)}\frac{q(\boldsymbol{x}_{t+1:T}|\boldsymbol{x}_t)}{q(\boldsymbol{x}_{t+1:T}|\boldsymbol{x}_t)} \\
&= p(c=0)\frac{p(\boldsymbol{x}_t|c=0)}{p(\boldsymbol{x}_t)} \\
&= p(c=0|\boldsymbol{x}_t)
\end{aligned}
\tag{20}
$$

## D    EXPERIMENTAL DETAILS

### D.1    IMPACT OF THE DIFFERENT HYPERPARAMETERS OF DNG

The hyperparameters introduced in the framework of DNG, being the prior $p(c)$, the temperature $\tau$ and the bias $\delta$ all have distinct effects. The prior dictates the initial guess for the posterior, i.e. $p_T(c|\boldsymbol{x}_T) = p(c)$ and therefore also the initial guidance scale $\lambda(\boldsymbol{x},T) = \lambda\frac{p(c)}{1-p(c)}$. Choosing a low prior can ensure that negative guidance is not immediately active. The temperature hyperparameter $\tau$ can help to reduce the fluctuations caused by the stochastic denoising process, we find that a value of around $\tau = 0.2$ works well in practice. The offset hyperparameter $\delta$ gives the model a slight bias towards increasing the prior, it could be alternatively described as making the prior time dependent. From a practical point of view, it's consequence is that when both the positively and negatively prompted models give similar prediction, the posterior increases faster, a highly desirable property. In practice, we suggest either choosing a relatively large prior with no offset (such as we have done for the MNIST experiments), or a low prior with a small offset (such as we have done for the CIFAR10 and Stable Diffusion experiments). We find that choosing an offset two, or even three, orders of magnitude lower than the temperature delivers satisfactory results.

### D.2    CLASS REMOVAL EXPERIMENTS

For the class removal experiments, unconditional MNIST and CIFAR10 diffusion models are required. For MNIST our own model is trained, while for CIFAR10 the pretrained model from Ho et al. (2020) is used[6]. For both datasets, a model trained on solely one of the classes is required. For this, our own models are trained on all the *zeros* of MNIST and all the *airplanes* of CIFAR10. Notice that all the guidance approaches are compared using the same two networks, reducing any additional bias due to the choices of network architectures. To analyze the generated images, a vision classifier is required. For MNIST our own basic convolutional based classifier is trained, obtaining over 98% accuracy over a test set. For CIFAR10, a pretrained vision transformer classifier is used[7] (Wu et al., 2020).

Hyperparameter values for our Negative Guidance scheme for the different datasets are given in Table 1. A discussion explaining the choice of the different hyperparameters of our scheme is included in D.1.

For Safe Latent Diffusion (Schramowski et al., 2023) a hyperparameter search was performed to obtain the values that perform best at high safety. The most important hyperparameter is the threshold value at which guidance is activated. We found that even in the setting of MNIST and CIFAR (quite far from the setting of Stable Diffusion in which the scheme is proposed), a threshold value of $\lambda_{\text{thresh}} = 0.04$ still performs optimally at high safety. This displays the flexibility of the approach proposed by Schramowski et al. (2023). The other hyperparameters are chosen as follows: $s_s = 100$, $\beta_m = 0.2$, $s_m = 0.1$. The meaning and impact of different hyperparameters is discussed in Appendix E. The absence of the offset for the MNIST experiments is compensated by choosing a much higher prior. The negative guidance scale has to be chosen significantly differently for the

---

[6]The pretrained model can be downloaded from huggingface at `https://huggingface.co/google/ddpm-cifar10-32`

[7]The pretrained classifier can be downloaded from Hugging Face at `https://huggingface.co/aaraki/vit-base-patch16-224-in21k-finetuned-cifar10`

| Dataset | Prior $p(c)$ | Temperature $\tau$ | Offset $\delta$ |
|---------|--------------|--------------------|-----------------|
| MNIST | 0.25 | 0.25 | 0.0 |
| CIFAR10 | 0.01 | 0.2 | 0.0002 |

Table 1: The prompt specific hyperparameters chosen for our Dynamic Negative Prompting.

various approaches. While in SLD the guidance scale can only be smaller or equal to the initial guidance scale $\lambda_0$, our scheme considers a dynamic self-regulating guidance, which can therefore require very different initial values. In practice, the guidance scale used in DNG is chosen one order of magnitude larger than that of NP. For NP or SLD, we chose values of around $0.5$, while for DNG we chose values around $5$.

### D.3 Text-to-Image experiments

To evaluate different negative guidance schemes in the context of T2I, we design 5 prompts that have a high likelihood of generating objects not included in the prompts themselves. The positive prompts generating these subjects are generated using ChatGPT[8]. For each prompt, a related and unrelated negative prompt is designed. The related negative prompt contains the name of the implicitly generated object, suggested by ChatGPT alongside the positive prompt. The semantically unrelated prompt is freely chosen by the authors. These prompts are given in Table 2. By generating small batches of images, we verified by hand that the implicitly generated objects are present when using the positive prompts. To allow the reader to analyze the prompts separately, these are visualized using different markers in Fig. 6.

We observe that the progressive generation of the various prompts can show diverse behaviors, further highlighting the complexity of a guidance in the context of Text-To-Image. This is discussed in more detail in Appendix F. The presented negative guidance results rely on prompt specific hyperparameters chosen in a limited interval and are all given in Table 3. It should be noted that different prompts might require different hyperparameters depending on the size and importance of the to remove object. Further work is required to obtain a general scheme that is effective in all circumstances. Developing this requires a substantially deeper understanding of the effect of dynamic guidance schemes, something also left for future work.

## E Safe Latent Diffusion: Details

Due to their non-constant guidance scale, Safe Latent Diffusion as introduced by Schramowski et al. (2023), is the most similar concurrent work present in the literature. Similarly to ours, their guidance scheme is only active in the case that the scores of the negative prompt and positive prompt are aligned. To make the comparison more apparent, their proposed pixelwise guidance scale can be rewritten in our notation as:

$$\mathbf{\Lambda}(\boldsymbol{x}, t) = \begin{cases} \lambda_0 \min\left(1, s_g \cdot \mid \boldsymbol{\epsilon}_{\boldsymbol{\theta},t} \ominus \boldsymbol{\epsilon}_{\boldsymbol{\theta},c\text{-},t} \mid \right) & \text{if } \boldsymbol{\epsilon}_{\boldsymbol{\theta},t} \ominus \boldsymbol{\epsilon}_{\boldsymbol{\theta},c\text{-},t} < \lambda_{\text{thresh}} \\ 0 & \text{else} \end{cases} \tag{21}$$

Notice that an important difference between the two approaches is the present of a pixel-wise varying guidance scale in SLD, highlighted by the matrix $\mathbf{\Lambda}(\boldsymbol{x}, t)$ versus our global, scalar guidance scale $\lambda(\boldsymbol{x}, t)$. Schramowski et al. (2023) also introduce additional momentum to the scheme, such that guidance can remain active for a limited time even if the score difference should increase a bit. They also add a late initialization of their approach. This is to specifically target the region of semantic generation with their guidance scheme.

In SLD the guidance scale, defined by Eq.(21), is always smaller than the initial guidance scale, which is not the case in our approach. Another difference, is that the metric they use to activate the negative guidance is a simple noise prediction comparison, very different from our dynamic, time dependent scheme.

Using different values for the threshold hyperparameter $\lambda_{\text{thresh}}$, as suggested by Schramowski et al. (2023), different safety regimes are obtained. All different proposed settings were tested, in the main

---

[8]Accessed before 30/09/2024

| Number | Name | Positive Prompt | **Related** NP | **Unrelated** NP |
|---|---|---|---|---|
| 1 | "Medieval feast" | "A grand medieval feast set in a great hall, filled with long tables covered in bountiful platters of food, goblets, and flickering light, with knights and nobles enjoying the lavish spread." | "Chalices, candles" | "Kids playing football" |
| 2 | "An English breakfast" | "A classic British breakfast, featuring a diverse selection of cooked delights, perfect for a filling and flavorful start to the day." | "Egg, sausage" | "The view of a skyline" |
| 3 | "A dinner table" | "A beautifully set dinner table, with elegant arrangements, a variety of enticing dishes, and delicate decor that speaks to the sophistication of the meal." | "Flowers, wine glasses" | "A person wearing sunglasses" |
| 4 | "An art workshop" | "An inspiring art workshop filled with creativity, featuring a wide range of tools and materials used by participants working on their individual projects." | "Paintbrush, canvases" | "A bicycle in the rain" |
| 5 | "An antique store" | "A charming antique store with shelves and tables filled with unique and historical treasures, offering a glimpse into the past." | "Lamps, old books" | "An electric toothbrush" |

Table 2: The five positive prompts with their respective related and unrelated negative prompts used for the experiments on Stable Diffusion. The related negative prompts are hand chosen after observing images generated using the positive prompts.

| Number | Name | Prior $p(c)$ | Temperature $\tau$ | Offset $\delta$ |
|---|---|---|---|---|
| 1 | "Medieval feast" | 0.01 | 0.2 | 0.004 |
| 2 | "A dinner table" | 0.01 | 0.3 | 0.003 |
| 3 | "An art workshop" | 0.01 | 0.2 | 0.002 |
| 4 | "An antique store" | 0.01 | 0.2 | 0.004 |
| 5 | "An English breakfast" | 0.01 | 0.3 | 0.003 |

Table 3: The prompt specific hyperparameters chosen for our Dynamic Negative Prompting.

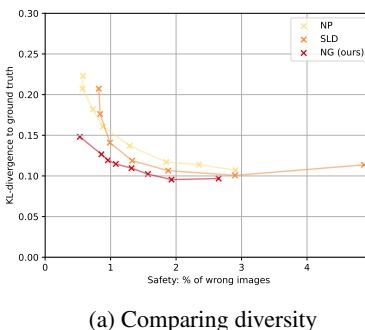 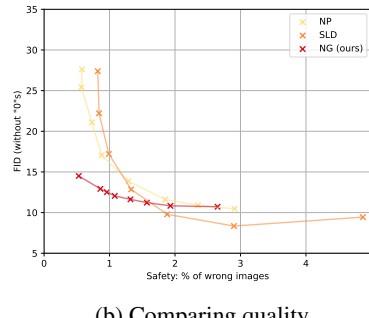

(a) Comparing diversity          (b) Comparing quality

Figure 10: Comparison of diversity and quality as a function of safety for SLD, NP and DNG (ours) when removing the class 0 ("airplanes") on CIFAR. The SLD hyperparameters are chosen to be least invasive, explaining the improved quality and relatively low safety of the approach.

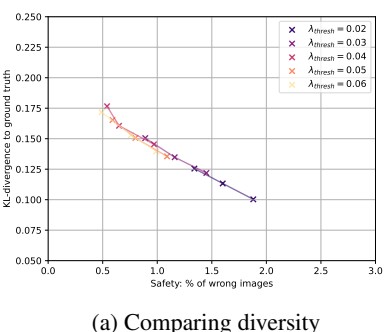 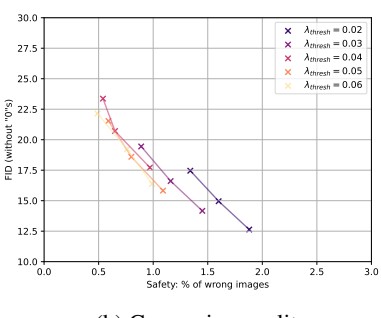

(a) Comparing diversity          (b) Comparing quality

Figure 11: Grid search over the threshold parameter of SLD $\lambda_{\text{thresh}}$ at high safety.

paper we show the curves containing the version that performs best at high safety, corresponding with the safest model. An interesting observation, visible in Fig. 10, is that by choosing a low threshold, one can obtain better FIDs with SLD than with rival approaches at low safety levels. We hypothesize that this is thanks to the small invasiveness of the SLD approach, which is mainly obtained thanks to its pixelwise guidance scale. This allows SLD to *locally* remove details from images, which is a major difference from both NP and our proposed DNG scheme. Notice that this superior performance is only at a very low safety level, for instance when there are still 3% of the original 10% of the forbidden class being generated.

A limited grid search over the threshold parameter of SLD $\lambda_{\text{thresh}}$ is also performed in the high safety regime and is shown in Fig. 11. As outlined by the authors, it is expected that when chosen too small the method is unsafe (i.e. still generates the unwanted features), and when chosen too high the method becomes equivalent to NP. We find these results still hold in the class-removal setting. An optimum is found around the range of $\lambda_{\text{thresh}} = 0.04$, which is the value initially proposed by the authors in the context of T2I and is therefore also the one used in this paper.

## F    STABLE DIFFUSION: EMPIRICAL OBSERVATIONS

As already highlighted in the literature (Kynkäänniemi et al., 2024; Raya & Ambrogioni, 2024; Brack et al., 2023; Sclocchi et al., 2024), the generation of images happens in phases. At high noise regime, low frequency details, such as large shapes or main colors, are generated. In the middle of the denoising process, the semantic content is generated. Towards the end of the process, when only little noise is remaining, the high-frequency fine details are generated. While these general observed truths are widely accepted, we observe that the specific location of these events is not an intrinsic property of the total T2I model, but instead also strongly depends on how the model is prompted. The main advantage of our self-regulating free guidance scale is that such events can be observed. By tracking the posterior, the model tells us itself when a specific feature is being generated.

When plotting the average guidance scale for the different prompts, it can be observed that these events are located at distinct moments in the diffusion process. The evolution of the guidance scale as a function of time is shown for the five prompts in Fig. 12. The posterior is orders of magnitude smaller in the case of the unrelated prompt, illustrating once more that our DNG can deactivate itself. The detailed analysis of the dynamic guidance scale in the case of the *related* negative prompts is left for future work. While the dynamic aspect of our guidance scale proves

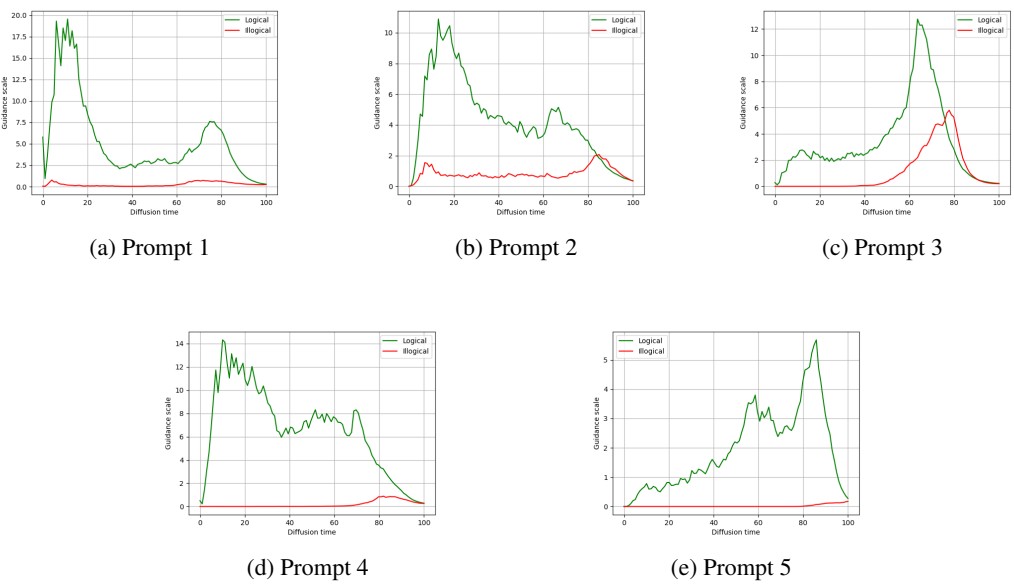

Figure 12: Evolution of the guidance scale throughout the diffusion process. Guidance scale averaged over 32 images. Maximal noise level is to the right of the x-axis. The guidance scale is respectively plotted in green and red in the cases of *related* and *unrelated* negative prompts

useful in the described scenario, one should be aware that it implies, up to a certain degree, a loss of user control. The self-regulating guidance is able to increase and decrease throughout the denoising process, which even though highly desirable, can also lead to unpredictable results when suboptimally tuned. While using NP increasing the guidance scale always results in stronger deviations from the unguided images, this does not have to be the case with our DNG scheme. It is perfectly possible that the guidance scale first decreases slightly and then follows the exact same trend, even if initialized higher. Losing this control is not problematic if tuned correctly, such as our scheme on MNIST or CIFAR10, as in that case a self-regulating dynamic guidance scheme is optimal. However, the effects of different parameters, as well as the prompt dependence of the scheme, need to be further studied to obtain a method working on large scales in the case of T2I.

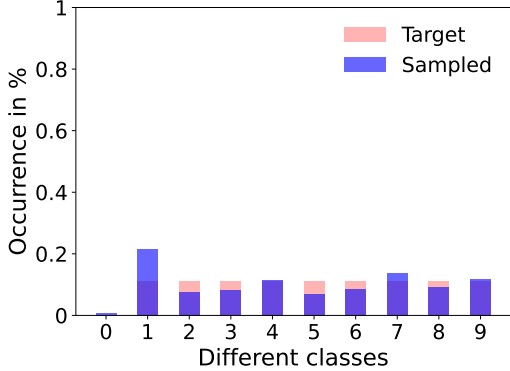

Figure 13: Visualization of how the KL-divergence is computed. In this example, the sampled distribution is obtained using DNG to remove "*0*" on MNIST. The classes "*1*" and "*7*" that are most different from the zero are oversampled, while similar classes such as the "*2*" or the "*5*" are undersampled.

## G    CLASS-REMOVAL: ADDITIONAL FIGURES

To demonstrate the robustness of our approach, the class removal experiments described in section 4.1 are repeated on four classes on both MNIST and CIFAR10[9]. For both datasets the first four classes are chosen. For MNIST this corresponds to the digits:"*0*"-'*3*", while for CIFAR10 this corresponds to the classes: "*airplanes*", "*cars*", "*birds*" and "*cats*"[10]. DNG outperforms concurrent approaches on the removal of all classes separately, highlighting its generalizability as well as its robustness. While the removal of all classes follow similar trends, such as an increase of FID as the safety increases, some specificities are different. We find the effect of negative prompting to be unreliable, meaning that removing different classes can result in large differences of performance. This is less the case for both SLD and DNG. The improved performance of DNG at high safety levels is especially visible when comparing different approaches using the FID metric.

---

[9]This will be extended to five classes before the end of the rebuttal period.

[10]The present graphs will be replaced by ones containing a larger number of points before the end of the rebuttal period. The class removal experiments on the number "*4*" as well as the class "*deer*" will by then also be added.

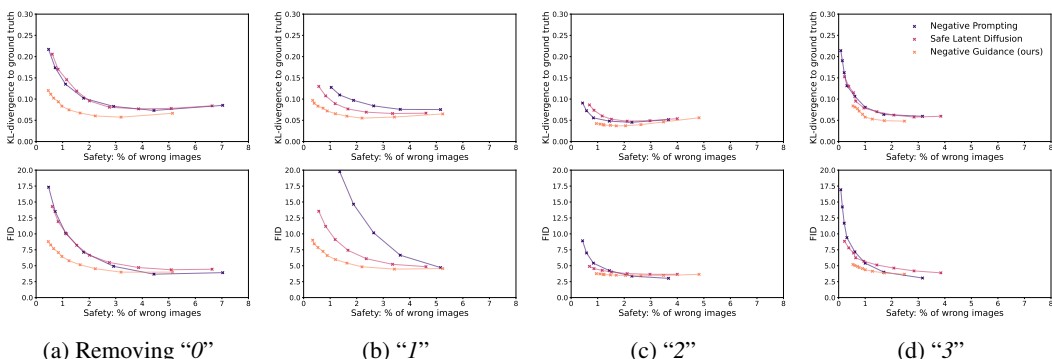

| (a) Removing "*0*" | (b) "*1*" | (c) "*2*" | (d) "*3*" |

Figure 14: Evaluation of the KL-divergence (top) and FID (bottom) on single class removal experiments performed on MNIST using Negative Prompting, Safe Latent Diffusion and our Dynamic Negative Guidance. Despite differences between classes, DNG outperforms both NP and SLD in all settings.

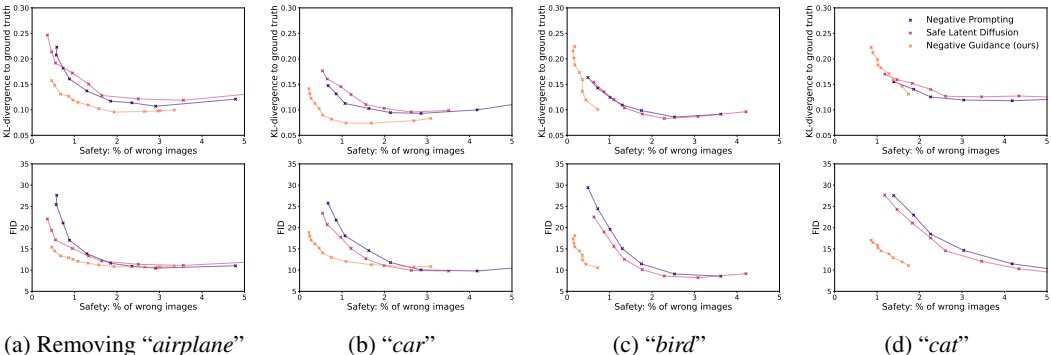

(a) Removing "*airplane*"    (b) "*car*"    (c) "*bird*"    (d) "*cat*"

Figure 15: Evaluation of the KL-divergence (top) and FID (bottom) on single class removal experiments performed on CIFAR10 using Negative Prompting, Safe Latent Diffusion and our Dynamic Negative Guidance. Despite differences between classes, DNG outperforms both NP and SLD in all settings.

## H  STABLE DIFFUSION: ADDITIONAL FIGURES

To visually compare how Negative Prompting and Dynamic Negative Guidance fare, we show some illustrative examples using the five prompts introduced in Appendix D. Results are compared by choosing the guidance scale such that the average cosine similarity of the *related* prompt is similar in both approaches. The initial guidance scale values used for the comparison for each prompt ($\lambda_{\text{NP}}$ and $\lambda_{\text{DNG}}$ for NP and DNG respectively) are written in the caption of Figures 16-20. These two, while serving the same purpose, are not one-to-one comparable for both schemes. A more thorough description of the connection between the two is done in Appendix D.1. In the case of unrelated guidance (orange box) as litlle change as possible is desired. This is often achieved using DNG (bottom row). On the other hand when using a related negative prompt (green box), changes in the image are expected. This is observed for both NP and DNG. The point of this setting is to be able to compare both approaches on similar global changes, meaning that both NP and DNG should alter the images in a similar fashion.

To provide more insights into how negative guidance takes effect, it is insightful to show what happens when the guidance scale is progressively increased. We display this in Figures 21-23.

To quantitatively compare DNG to NP, we propose a new metric that consists of a combination of the average CLIP scores measured in the unrelated and related prompted cases. The former should be as large as possible (i.e. as close to one as possible), implying that unrelated negative prompts have a negligible effect. On the other hand the CLIP scores computed using a related negative prompt should significantly differ from one, illustrating that the original image has been altered. The proposed metric $Q$ consists of the difference between the average CLIP Scores in the unrelated and related cases, or thus:

$$Q(\text{img}_{\text{no guid}}, \text{img}_{\text{rel.}}, \text{img}_{\text{unrel.}}) = \text{CS}(\text{img}_{\text{no guid}}, \text{img}_{\text{unrel.}}) - \text{CS}(\text{img}_{\text{no guid}}, \text{img}_{\text{rel.}}) \quad (22)$$

Thanks to their linear form clearly visible in Fig. 6b the average CLIP-scores can be used to obtain a single scalar value. With as linear regression parameters $a, b$ of the linear function visible in Fig. 6b and $[x_{\min}, x_{\max}]$ the range for which CLIP score in the related case are obtained, the metric can be computed as:

$$\begin{aligned} Q(\text{img}_{\text{no guid}}, \text{img}_{\text{rel.}}, \text{img}_{\text{unrel.}}) &= a\frac{x_{\max} + x_{\min}}{2} + b - \frac{x_{\max} + x_{\min}}{2} \\ &= (a-1)\frac{x_{\max} + x_{\min}}{2} + b \end{aligned} \quad (23)$$

This value can be visually interpreted as the average $y$ value from which the average $x$ value is substracted for every set of points in Fig. 6b. The values of our metric $Q$ computed on the sets of points obtained using both NP and DNG are given in table 4, in which DNG outperforms NP, corresponding with higher $Q$ values.

| Prompt Number | 1 | 2 | 3 | 4 | 5 | Average |
|---|---|---|---|---|---|---|
| NP | -0.015 | 0.003 | 0.017 | 0.016 | 0.041 | $0.007 \pm 0.014$ |
| DNG (ours) | 0.053 | 0.018 | 0.080 | 0.054 | 0.057 | $0.052 \pm 0.022$ |

Table 4: The values of the $Q$ metric as defined in eq. (23). Higher values correspond to improved performance.

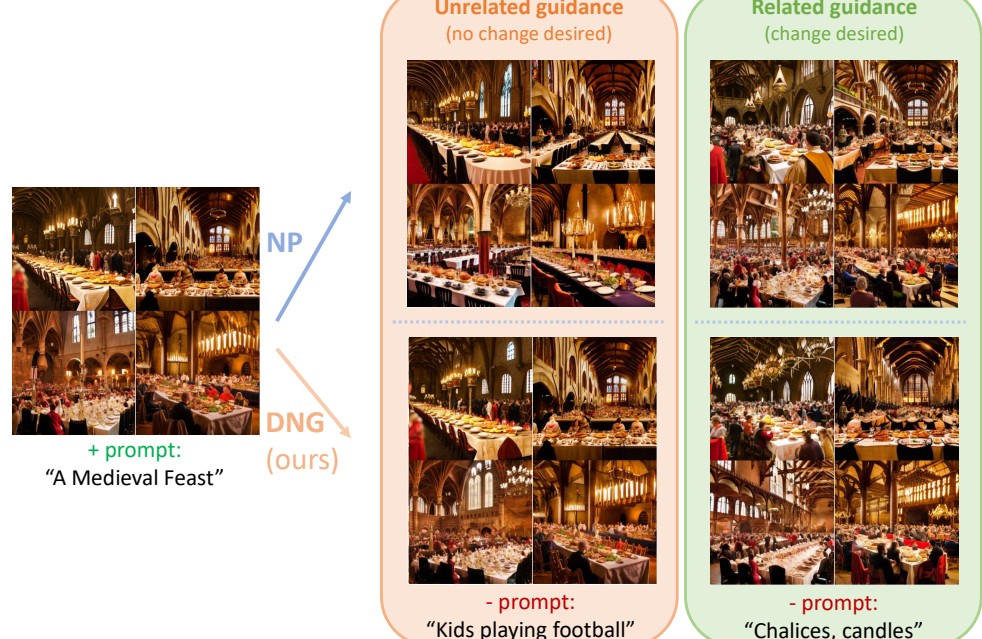

Figure 16: Examples comparing NP (up) to our DNG (down) in the case of *unrelated* (left) and *related* (right) negative prompts for: **Prompt 1**. Guidance scales: $\lambda_{\text{NP}} = 3.4$, $\lambda_{\text{DNG}} = 27$. NP still alters the images using unrelated negative prompts, for instance it removes all people present in the unguided image. This most likely happens because the negative prompt contains the word "*Kids*".

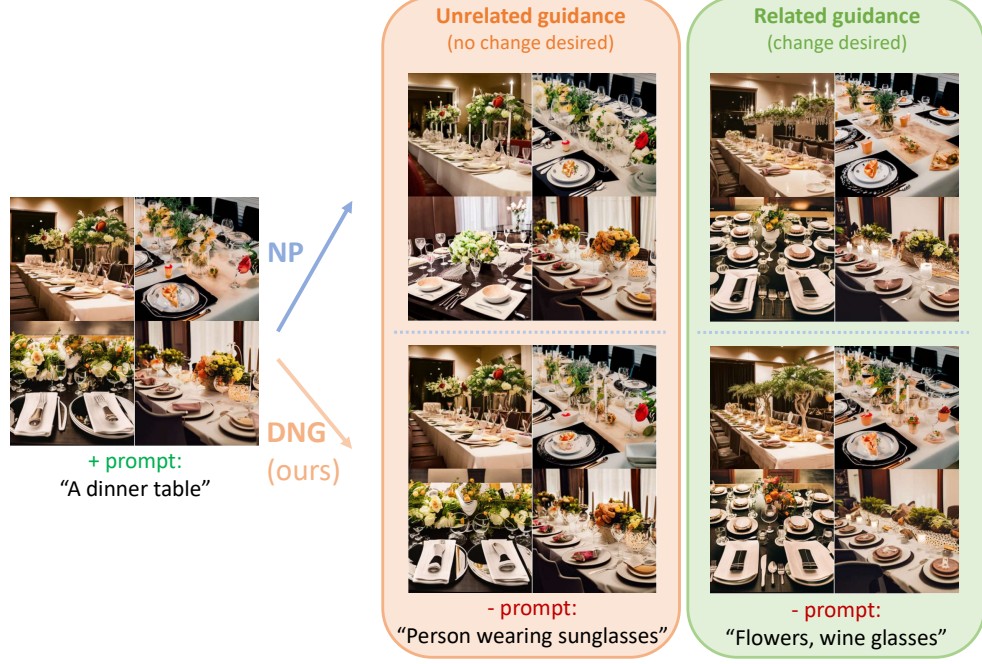

Figure 17: Examples comparing NP (up) to our DNG (down) in the case of *unrelated* (left) and *related* (right) negative prompts for: **Prompt 2**. Guidance scales: $\lambda_{\text{NP}} = 1.7$, $\lambda_{\text{DNG}} = 12$. In contrast to DNG, NP alters images using unrelated negative prompts (see for instance the bottom left image), while both NP and DNG perform similarly at removing the flowers from the unguided image.

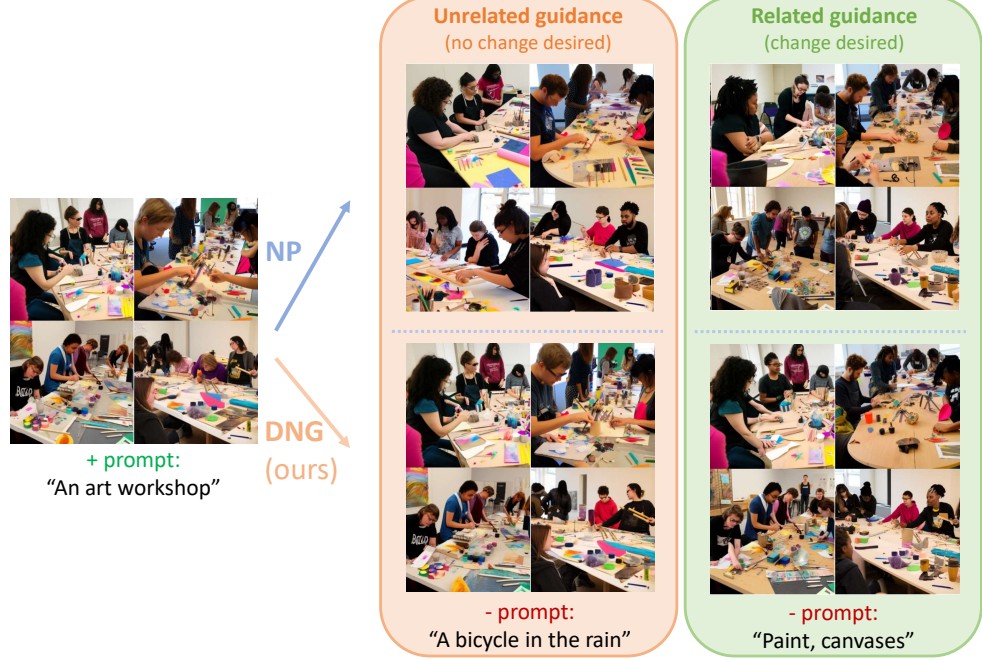

Figure 18: Examples comparing NP (up) to our DNG (down) in the case of *unrelated* (left) and *related* (right) negative prompts for: **Prompt 3**. Guidance scales: $\lambda_{\text{NP}} = 1.7$, $\lambda_{\text{DNG}} = 27$. NP alters images using unrelated negative prompts, while DNG leaves most of the images unaltered (see for instance the top right image). Both NP and DNG perform similarly in the case of the related negative prompt.

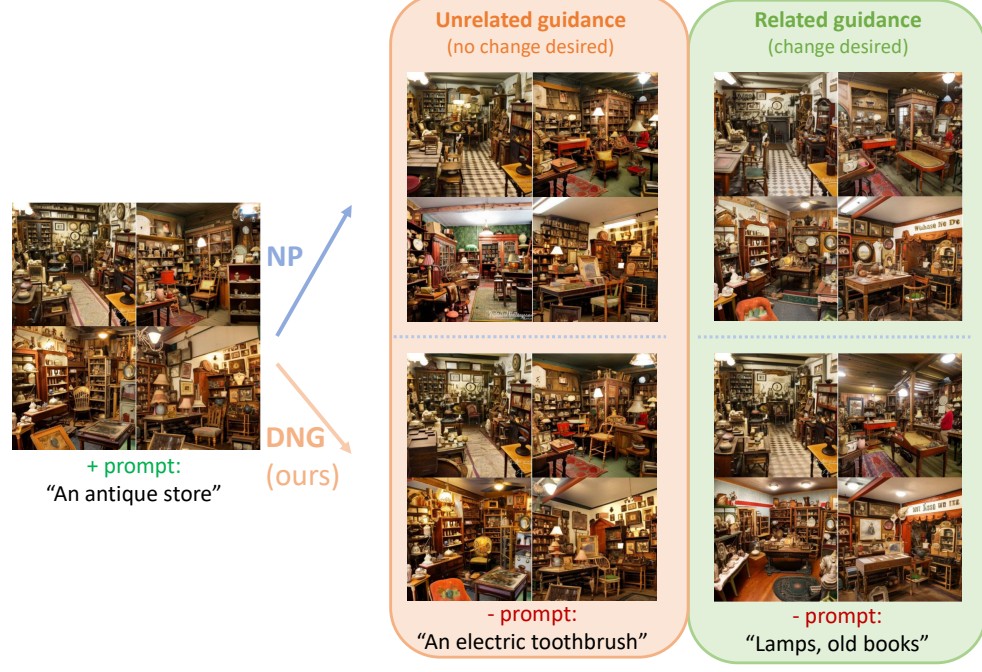

Figure 19: Examples comparing NP (up) to our DNG (down) in the case of *unrelated* (left) and *related* (right) negative prompts for: **Prompt 4**. Guidance scales: $\lambda_{\text{NP}} = 2.55$, $\lambda_{\text{DNG}} = 21$. NP alters images using unrelated negative prompts, while DNG leaves most of the images unaltered (see for instance the floor of the top left image).

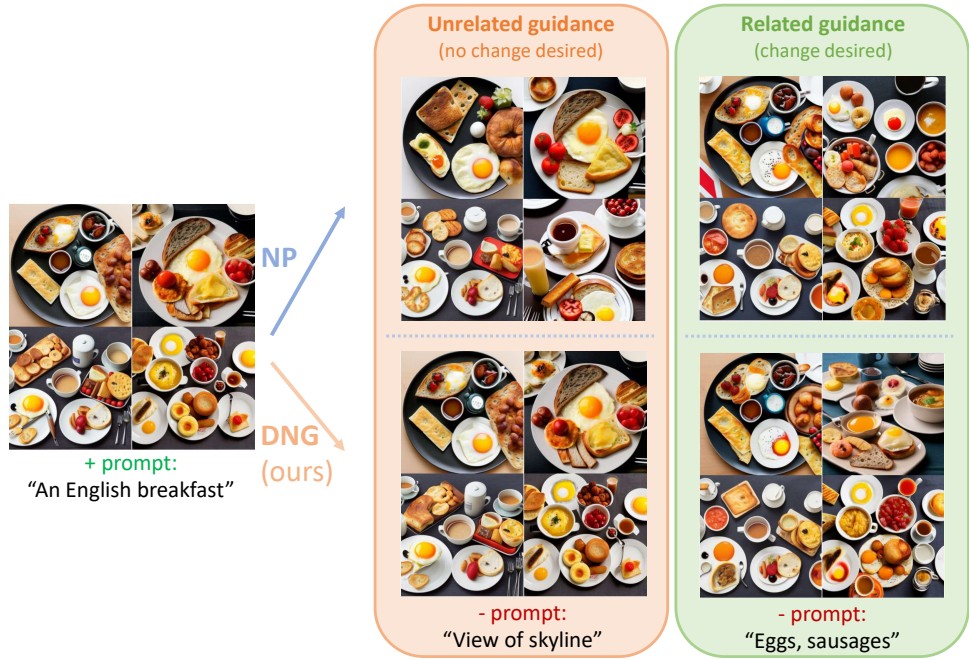

Figure 20: Examples comparing NP (up) to our DNG (down) in the case of *unrelated* (left) and *related* (right) negative prompts for: **Prompt 5**. Guidance scales: $\lambda_{\text{NP}} = 3.4$, $\lambda_{\text{DNG}} = 18$. NP alters images using unrelated negative prompts, while DNG leaves most of the images unaltered (see for instance the number of dishes in the bottom right image). Both NP and DNG perform similarly in the case of the related negative prompt (see for instance the removal of the egg in the bottom left image).

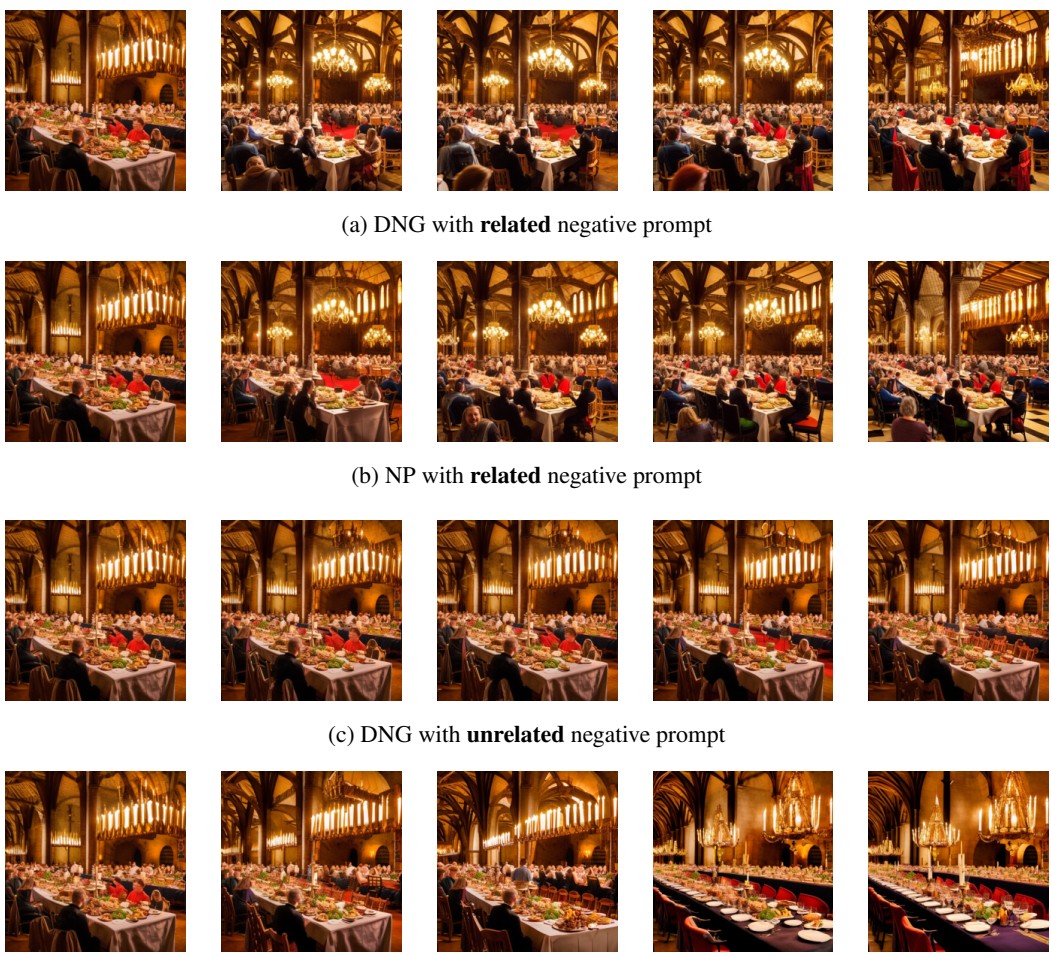

Figure 21: Progressive impact of negative prompts with increasing guidance scale from left to right. In (a) and (b) the **related** negative prompt is used, while in (c) and (d) the **unrelated** negative prompt is used. Analysis performed for: **Prompt 1**

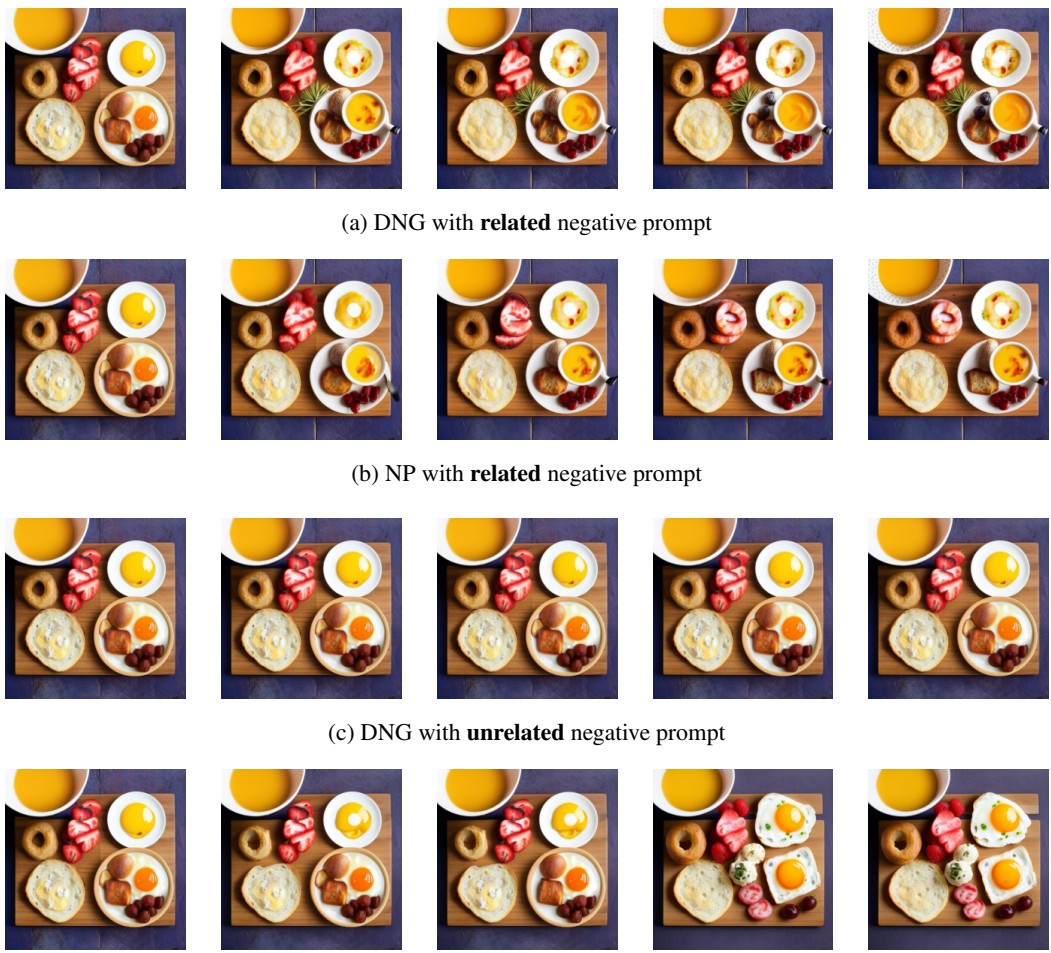

(a) DNG with **related** negative prompt

(b) NP with **related** negative prompt

(c) DNG with **unrelated** negative prompt

(d) NP with **unrelated** negative prompt

Figure 22: Progressive impact of negative prompts with increasing guidance scale from left to right. In (a) and (b) the **related** negative prompt is used, while in (c) and (d) the **unrelated** negative prompt is used. Analysis performed for: **Prompt 2**

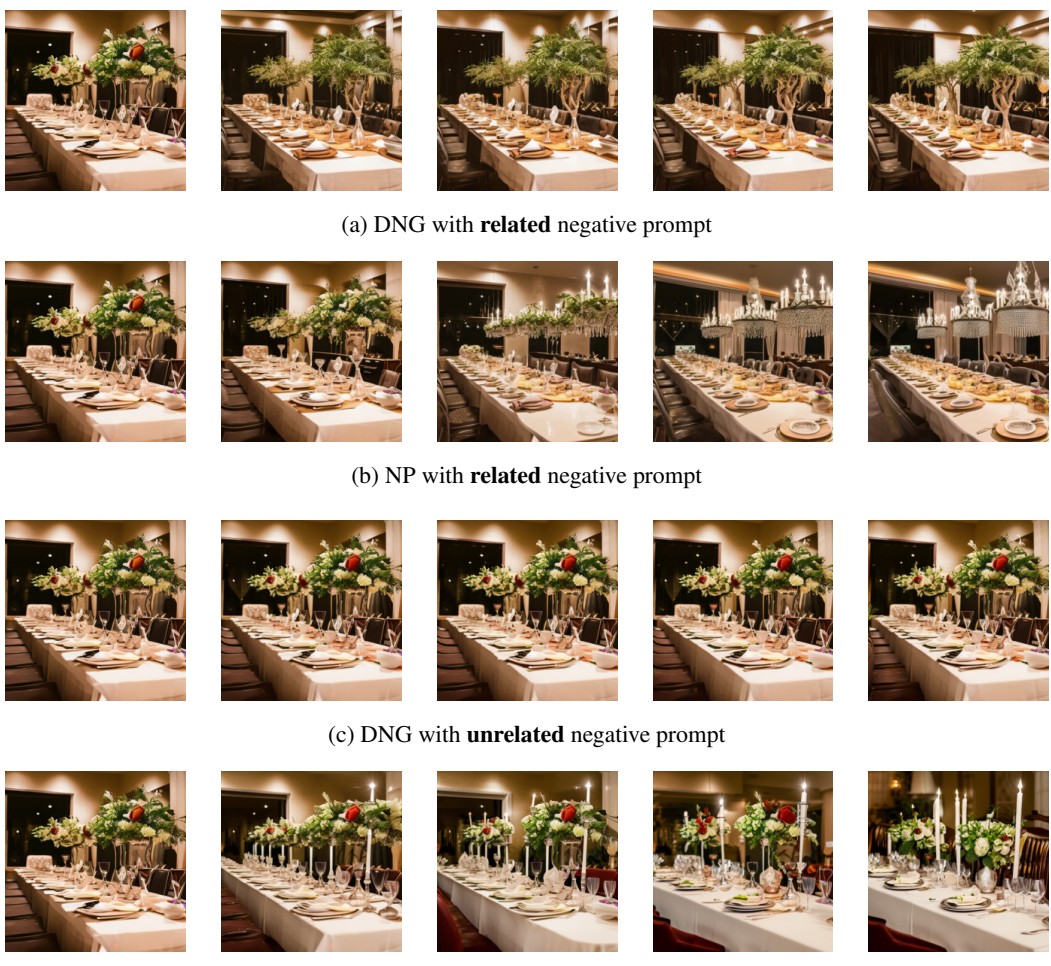

(a) DNG with **related** negative prompt

(b) NP with **related** negative prompt

(c) DNG with **unrelated** negative prompt

(d) NP with **unrelated** negative prompt

Figure 23: Progressive impact of negative prompts with increasing guidance scale from left to right. In (a) and (b) the **related** negative prompt is used, while in (c) and (d) the **unrelated** negative prompt is used. Analysis performed for: **Prompt 3**

