# OpenReview forum: "Dynamic Negative Guidance of Diffusion Models"
_ICLR.cc/2025/Conference — ICLR 2025 Poster_

### Official Review · Reviewer_JR45 · 2024-10-26

**Soundness:** 3
**Presentation:** 3
**Contribution:** 3
**Rating:** 6
**Confidence:** 4

**Summary:**

The paper presents Dynamic Negative Guidance (DNG), a novel technique for improving Negative Prompting (NP) in diffusion models, specifically addressing limitations in text-to-image applications. Conventional NP assumes a fixed guidance scale to suppress undesired features, which can result in poor performance due to the non-stationary and state-dependent nature of the reverse process in diffusion models. DNG overcomes this by introducing an adaptive approach that adjusts the guidance dynamically based on time and state, thereby refining the model’s ability to avoid generating unwanted features.

**Strengths:**

1. the DNG method in novel and has strong practical significance
2. paper did a comprehensive evaluation on safety, class balance, and image quality on MNIST and CIFAR10.
3. the paper is straightforward and easy to follow

**Weaknesses:**

1. terms like Dynamic Negative Guidance and guidance scale could better deserve a brief contextual note, as not all readers may be familiar with their meaning in this context.
2. the paper needs to avoid jargon without explanation.
3. the paper needs to improve flow and conciseness.

**Questions:**

see weakness

---

> ### Author Response · Authors · 2024-11-23
> **Response to Reviewer JR45**
>
> **Summary:**
>
> We would first like to thank the reviewer for their comments and are glad that they found the paper easy to follow and the method of strong practical significance.
> Below we address each the indicated weaknesses:
>
> **Weaknesses:**
>
> **W1)** Terms like Dynamic Negative Guidance and guidance scale should be explained:
>
> We have added an explanation for the term dynamic negative guidance scale and what it means in the context treated in the paper (see lines 249-252).
>
> **W2)** The paper needs to avoid jargon without explanation.
>
> We have gone through the paper in detail to avoid the use of unnecessary jargon (for example, we left out the allusion to force fields previously present in, for instance, section 3) and have carefully defined concepts that are essential to this work (such as the term ‘dynamic guidance’).
>
> **W3)** The paper needs to improve flow and conciseness.
>
> We believe that by having once again gone through the entirety of the paper and subtly modifying the text, we have improved both the flow and conciseness of the paper. In addition, the joint comments from all reviewers and our corresponding adaptations of the manuscript, have considerably increased the reader’s insights into the proposed method, even in the main body of the paper. For example, the addition of Fig. 5 (visualization of diffusion trajectories), and the more formal evaluation (CLIP scores in Fig. 7b) should also help the reader appreciate the novelty/usefulness of our dynamic negative guidance scale.

---

### Official Review · Reviewer_v6iu · 2024-10-27

**Soundness:** 3
**Presentation:** 3
**Contribution:** 3
**Rating:** 6
**Confidence:** 4

**Summary:**

In text-to-image generation, negative prompts are used to guide the generative model not to create something. This paper presents a method called dynamic negative guidance that aims to be safer, less invasive than regular negative prompting in guiding a T2I model to create desirable images while avoiding unwanted components. The dynamic negative guidance relies on a near-optimal time and state dependent modulation of the guidance without requiring additional training. The method has been tested in MNIST and CIFAR-10.

**Strengths:**

A strength of the proposed method, DNG, is to estimate the posterior by tracking the discrete Markov chain during the denoising process. The strength of the guidance is dynamically related to how close the negative prompt is related to the positive prompt. This seems to a strength of the method since it can adaptively determine whether the negative prompt is even relevant at all. To the contrary, existing negative prompting methods may not be able to ignore irrelevant negative prompts. The proposed method may overcome the weakness of existing NP methods that blankly try to invert the force field to move away from the positive prompts without precisely moving away from the negative prompts. This is reflected in the factor pt(c-|x)/(1−pt(c-|x)) in Eq 10.

**Weaknesses:**

Writing can be better and consistent. For example, the paper has both c_- and c-, it should be consistent.
The biggest weakness with the paper is the example images given at the end, starting from page 22. First, the pictures are so small, it is hard to appreciate the difference between NP results and DNG results. Second, I could be missing something, but it seems there is no big difference between the two types of results. In particular, it seems NP results are not bad or accidently include the undesirable features given in the negative prompts.
Similar comment for Figure 6, as it does not seem that NP results in the presence of negative prompt "view of skyline" is incorrect.

**Questions:**

While the mathematical derivation of DNG looks reasonable, the generated image examples are hard to understand.
Can authors point out the difference between NP results and DNG results in the example figures, such as Figure 6 and figures starting on page 22?
The purpose of experiments of Section 4.1 and corresponding Figure 5 are not clear. I don't understand why the experiments need to remove one class in MNIST and one class in CIFAR-10. If the purpose of the method is to avoid generating undesirable features, shouldn't the model be trained on all classes and only in practical use of the method, one can prompt the model to not generate something, for example, not to generate number zero or an airplane. Or am I missing something here.
While Table 2 lists the positive prompts and related and unrelated negative prompts, can authors give examples on how exactly a full prompt was written up in English and fed to a T2I model?
In Figure 10(b), as the authors explained, because SLD may have less invasiveness, it gave better FID than DNG, which is reasonable, but why from the figure it seems NP also had a lower FID than DNG as % of wrong images went up? Can authors elaborate on this?
From both Figure 10(a) and (b), it seems that NP had overall worse performance than SLD and DNG, but it also appears that NP performance in KL divergence and FID had a decreasing pattern while those of DNG appeared to pick up at higher % of wrong images, what could be the reason behind this observation? Can authors discuss about this?
Figure 11, for illogical prompts, the guidance scales were larger at later diffusion time than at early diffusion time, can authors provide intermediate generative results corresponding to this change of the guidance scale?

---

> ### Author Response · Authors · 2024-11-23
> **Response to Reviewer v6iu**
>
> **Summary:**
>
> We would first like to thank the reviewer for their thorough review of the manuscript. We greatly appreciate their recognition of the innovativeness/usefulness of the proposed posterior estimation. The comments/questions are highly valuable and have helped us to improve the document.
> Below we address each of the indicated weaknesses and questions:
>
> **Weaknesses:**
>
> **W1)** Consistency of notation:
>
> We have gone over the document and removed as many inconsistencies as possible (such as those previously present in section 3.2).
>
> **W2)** Overly small images:
>
> We agree with the reviewer that the illustrative images shown were too small. We have replaced the 4x4 grids of appendix H by 2x2 grids containing randomly sampled images.  It makes the paper much more visually attractive, thanks for this excellent suggestion.
>
> **W3)** Limited distinction between NP and DNG:
>
> The NP results are not erroneous (i.e. still follow the original prompt). These have however, in contrast to DNG, been altered from the unguided setting. The consequence of this is a loss of diversity, which may have undesired consequences.  For example, in the MNIST case, by removing the ‘0’ class, generation of instances of similar classes, such as ‘2’, become much less likely - see figure 12 in appendix G.
> It is our belief that a good negative guidance scheme should not only be safe, but also maximally preserve diversity. To clarify this key point, we have added a paragraph to the main document (see lines 468-71). We thank the reviewer for pointing out that the original document was not clear enough on the matter.
>
> **Questions:**
>
> **Q1)** Explain differences between NP and DNG results in the figures?
>
> We agree with the reviewer that such explanations were missing from the manuscript. These have been added in the main text (lines 466-468) and in the captions of the additional samples provided in appendix H. We thank the reviewer for mentioning this.
>
> **Q2)** Extend the single-class removal experiments to multiple classes?
>
> A conditional model would indeed serve just as well as class-specific models. To demonstrate the robustness and generalizability of our approach, we have extended our results to the removal of three other classes per dataset and show that DNG also outperforms concurrent approaches in these settings. The results obtained on different CIFAR classes are visible in Fig. 4 while those obtain on MNIST are visible in appendix G. We believe that these experiments significantly contribute to the quality of the evaluation of our approach.
>
> **Q3)** Elaborate on choice of prompts for T2I case?
>
> The main text has been adapted to better explain our choice of positive/negative prompts (see line 427). A paragraph explaining this in more detail has been added to appendix D.3.
>
> **Q4)** Add a comparison of FID scores on low safety?
>
> By extending the single class removal experiments to multiple classes, we observe that the FID of NP is not consistently lower than that of DNG at low safety regime (see the new Fig 4). It should also be noted that this regime is of limited practical use, as when generating, for instance 5% of forbidden images out of the original 10%, the negative guidance scheme can be considered practically ineffective. Therefore, the most relevant regime of a negative guidance scheme is that of high safety.
>
> **Q5)** Provide intermediate generative results?
>
> To further illustrate our dynamic guidance scheme, we have added a figure containing specific diffusion trajectories as well as their corresponding dynamic guidance scales to the main text (see Fig. 5 discussed in lines 429-463). We hope that these will provide the readers with additional insights into DNG.  We would like to thank the reviewer for this very valuable suggestion, which in our opinion nicely illustrates the strength and flexibility of our proposed approach.

---

> > ### Comment · Reviewer_v6iu · 2024-11-26
> >
> > I maintain the original score.

---

### Official Review · Reviewer_UeKs · 2024-11-01

**Soundness:** 1
**Presentation:** 2
**Contribution:** 2
**Rating:** 5
**Confidence:** 4

**Summary:**

The authors argue that conventional negative prompting methods are constrained by the assumption of a constant guidance scale. To address this limitation, they propose a novel dynamic negative guidance technique that adapts the guidance scale based on both time and state, aiming for near-optimal modulation. Notably, this approach does not require additional training. The authors evaluate their method on MNIST and CIFAR10, comparing it against various baselines, and demonstrate its effectiveness. They also show that the technique integrates well with Stable Diffusion, offering improved accuracy in defining negative prompts.

**Strengths:**

• The proposed method demonstrates promising results on MNIST and CIFAR10, outperforming standard negative prompting techniques and safe latent diffusion methods. This improvement suggests that the dynamic guidance approach offers a meaningful advantage in generating more accurate outputs for image datasets with complex features.

• Preliminary results with Stable Diffusion also appear promising, indicating that the method may effectively enhance prompt accuracy within more sophisticated generative models. However, additional evaluation would further substantiate these findings and provide more insight.

• Though a minor detail, the use of color highlights in algorithms and formulas is a thoughtful touch that enhances readability.

**Weaknesses:**

• There is a lot of research happening in the field of negative prompting, yet this paper lacks a comprehensive comparison with many  leading methods. An in-depth comparison would have more clearly illustrated the strengths and limitations of this approach relative to existing techniques, helping to clarify its unique contributions.

• Overall, the evaluation of the proposed method lacks depth. A more thorough and systematic assessment across various scenarios and metrics would strengthen the validity of the results and give a clearer picture of the method’s real-world applicability and robustness.

• As acknowledged by the authors, a significant limitation of this manuscript is the limited evaluation of text-to-image generation, which is the primary application area for this method. Without a comprehensive exploration of T2I use cases, the potential impact of this work is somewhat undermined, leaving much of its promise unexplored.

• Additionally, the absence of quantitative metrics is a notable gap. Deferring these metrics to future work is a missed opportunity, as they would have added rigor to the analysis and allowed for a more objective assessment of the method's effectiveness.

**Questions:**

Given that you acknowledge these limitations in the manuscript, could you clarify why they were not addressed in the current version? Including these aspects seems essential to strengthen the manuscript's rigor and completeness.

---

> ### Author Response · Authors · 2024-11-23
> **Response to Reviewer UeKs**
>
> **Summary:**
>
> We would like to thank the reviewer for their detailed analysis of the manuscript, which we believe will substantially improve the final work.  We appreciate that the author recognizes the effort we made to make the mathematical discussion as intuitive/readable as possible. We are grateful that the reviewer recognizes the improvements obtained by using our DNG scheme in the context of class-conditional generation.
> Below we address each of the indicated weaknesses and questions:
>
> **Weaknesses:**
>
> **W1)** Additional comparison with literature needed:
>
> We have added a paragraph discussing the current literature regarding Negative Prompting at the beginning of section 2.3 (lines 164-169). In particular we have chosen to focus on “Understanding the impact of Negative Prompts”, the “PerpNeg” algorithm as well as the already discussed Safe Latent Diffusion method.
>
> **W2)** The evaluation lacks depth. More systematic assessment needed.
>
> To emphasize the robustness of DNG, we have compared it to the baselines (SLD, NP) on the removal of three additional classes on MNIST and CIFAR10, added to the revised manuscript (additional simulations are still running). In all cases, DNG outperforms concurrent approaches. The results obtained on CIFAR are now visible in Fig. 4, while those obtained on MNIST can be found in appendix G. We believe that these strong results both highlight the robustness of DNG and display its generalizability to different image semantics. We thank the reviewer for the suggestion, as it definitely improves the thoroughness of our evaluation.
>
> **W3-4)** Limited evaluation of text-to-image generation and absence of quantitative metrics:
>
> The goal of the T2I results is not to show that DNG generates more qualitative images than NP, but to demonstrate that DNG better preserves image diversity thanks to its ability to deactivate itself in the case of unrelated negative prompts. To make this clearer, we propose replacing our latent space cos-sim metric by the more widespread CLIP-Score between the unguided and negatively guided images (see Fig. 7.b). We also add a table comparing the average CLIP-Score difference when guiding using NP vs. DNG to appendix H. While it is true that the FID metric could provide intuition into how the quality of the images is preserved using various methods, this would require a much larger prompt dataset containing both positive prompts and a representative set of associated negative prompts, which is hard to compose.
>
> **Questions:**
>
> **Q1)** Limitations are indicated in the manuscript, why are they not directly addressed?
>
> We believe that the more rigorous analysis of DNG on class conditional generation (removing multiple classes in the revised version, instead of a single one in the original manuscript) better demonstrates the robustness and generalizability of DNG.  The introduction of CLIP-Score metric in the context of T2I further demonstrates that even in the case of complex image semantics, DNG can preserve the diversity of the underlying model, which we argue to be a valuable asset.

---

> > ### Comment · Reviewer_UeKs · 2024-11-27
> >
> > Thank you for the clarifications. I remain my original score

---

### Official Review · Reviewer_xP5A · 2024-11-02

**Soundness:** 3
**Presentation:** 3
**Contribution:** 3
**Rating:** 8
**Confidence:** 4

**Summary:**

In this paper, a new concept called Dynamic Negative Guidance (DNG) is proposed, which is an improvement on the existing Negative Prompting (NP) method in diffusion models.

**Strengths:**

1. By dynamically adjusting the intensity of negative prompts, DNG solves the problem that the traditional NP method may encounter suboptimal results or complete failures in the generation process.
2. The structure of the paper is clear and the content is well organized.
3. DNG combines the needs of diffusion model (DMs) and condition generation, and estimates the posterior probability by tracking discrete Markov chains in the generation process. This method is an innovative extension of the existing technology.

**Weaknesses:**

1. Although the paper has been experimentally verified on MNIST and CIFAR10 datasets, the performance of DNG for more complex tasks (Text to image) has not been fully verified.
2. DNG is compared with NP and SLD methods in this paper, but the comparison of each method may lack in-depth analysis, especially the performance comparison under different parameter settings.

**Questions:**

1. You have demonstrated the effectiveness of DNG on the MNIST and CIFAR10 datasets. How does DNG perform on more complex and diverse data sets, such as ImageNet, especially on different image semantics and complexity?
2. The paper contains some visualizations, but can you provide more detailed visualizations to show the progressive impact of DNG in the image generation process, especially how it dynamically adjusts in the denoising step?

---

> ### Author Response · Authors · 2024-11-23
> **Response to Reviewer xP5A**
>
> **Summary:**
>
> We would first like to thank the reviewer for their useful feedback and are happy that they found the paper clear and well organized. Many thanks also for noting the innovativeness of the posterior estimation through the Markov Chain.
> Below we address each the indicated weaknesses and questions:
>
>
> **Weaknesses:**
>
> **W1)** Not fully verified on more complex tasks:
>
> We acknowledge that the analysis of DNG in the text-to-image setting is limited. To improve the evaluation of the Stable Diffusion experiments, we applied the established CLIP embedding based cosine similarity.   Also see our answer to question Q1 below.
>
> **W2)** Performance comparison under different parameter settings lacks depth:
>
> The hyperparameters for SLD were chosen by tuning the hyperparameters (specifically the threshold parameter) on each specific dataset. To satisfy possible concerns we are currently running a more exhaustive grid search for high safety regimes which will be added to appendix E before the end of the rebuttal period. We will add a figure similar to Fig. 4, but restricted to high safety regimes (under 2%) and containing various settings of SLD. NP does not have any other hyperparameters except the guidance scale over which the sweep is performed.
>
>
> **Questions:**
>
> **Q1)** Performance of DNG on more complex and diverse data sets?
>
> To analyze the performance of DNG on different image semantics we have repeated the single class removal experiments on 4 classes for both MNIST and CIFAR. The results on the various classes of CIFAR have been added to the main document (shown in Fig. 4), while the results on MNIST have been added to appendix G. Similarly to the two classes originally shown, DNG surpasses concurrent approaches in all settings. These results further demonstrate the robustness of our approach.
> The experiments on Stable Diffusion with a more formal evaluation (CLIP scores) further highlight the promise of DNG on more complex image datasets
>
> **Q2)** More visualizations of DNG in the image generation process?
>
> To help the reader appreciate the dynamics of our approach, we have added a figure containing two diffusion trajectories in the context of T2I to the main text (Fig 6, described in lines 429-63). The diffusion trajectories are accompanied by a plot of their dynamic negative guidance scale. We thank the reviewer for this valuable suggestion, as we believe it really adds to the story of the paper and leads to improved insights in our proposed method.

---

> > ### Comment · Reviewer_xP5A · 2024-11-27
> >
> > I maintain the original score.

---

### Author Response · Authors · 2024-11-28
**Summary of reviews and changes made during rebuttal**

We would like to thank the reviewers for their detailed feedback, which has helped us significantly improve the quality of the manuscript.

The reviewers agreed both on the innovativeness and the practical significance of the posterior estimation scheme obtained by tracking the Markov Chain. The reviewers also all found the paper clear and well-organized.

They argued that generalizability of the results could have been better demonstrated.  In response, we extended the class removal experiments on CIFAR and MNIST from one to four classes (see Fig. 4), and the reported results confirm the robustness of our dynamic negative guidance scheme.

Some concerns were expressed over the choices of hyperparameters for concurrent baseline approaches, in particular that of safe latent diffusion (SLD).  In response, we performed a sweep over the SLD threshold parameter, confirming that the choice already proposed by the authors in the context of T2I remains applicable in the class removal setting.

Furthermore, in answer to some reviewers' suggestion to make the dynamic negative guidance scheme better understandable, we have added a new figure (Fig. 5) showing diffusion trajectories with their according dynamic guidance scales. We believe that these figures both provide great insight and demonstrate the relevance of our scheme. We thank the reviewers for this valuable suggestion.

---

### Meta-Review · Area_Chair_NHdp · 2024-12-19

**Metareview:**

This paper proposed dynamic negative guidance for diffusion models instead of previous negative prompt with a constant scale. Overall, the paper is technically sound. However there are also some issues with the paper, such as the lack of validation in more complex and diverse applications, lack of depth in evaluation, etc. While the reviewers maintain their scores after rebuttal, the responses seem to address these concerns partially. Given that the authors need some space for the theoretical analysis of the problem, I believe it is hard for the authors to include a very diverse evaluation in the results. Nevertheless, this is a conference that has very limited space. I would suggest the authors to further validate the work in more applications in the future works.

**Additional Comments On Reviewer Discussion:**

NA

---

### Decision · Program_Chairs · 2025-01-22

Accept (Poster)